EMBO
Molecular Medicine

# Lactation opposes pappalysin-1-driven pregnancy-associated breast cancer

Yukie Takabatake[1], Claus Oxvig[2], Chandandeep Nagi[3], Kerin Adelson[4], Shabnam Jaffer[3], Hank Schmidt[4], Patricia J Keely[5], Kevin W Eliceiri[6], John Mandeli[7] & Doris Germain[1,*]

## Abstract

Pregnancy is associated with a transient increase in risk for breast cancer. However, the mechanism underlying pregnancy-associated breast cancer (PABC) is poorly understood. Here, we identify the protease pappalysin-1 (PAPP-A) as a pregnancy-dependent oncogene. Transgenic expression of PAPP-A in the mouse mammary gland during pregnancy and involution promotes the deposition of collagen. We demonstrate that collagen facilitates the proteolysis of IGFBP-4 and IGFBP-5 by PAPP-A, resulting in increased proliferative signaling during gestation and a delayed involution. However, while studying the effect of lactation, we found that although PAPP-A transgenic mice lactating for an extended period of time do not develop mammary tumors, those that lactate for a short period develop mammary tumors characterized by a tumor-associated collagen signature (TACS-3). Mechanistically, we found that the protective effect of lactation is associated with the expression of inhibitors of PAPP-A, STC1, and STC2. Collectively, these results identify PAPP-A as a pregnancy-dependent oncogene while also showing that extended lactation is protective against PAPP-A-mediated carcinogenesis. Our results offer the first mechanism that explains the link between breast cancer, pregnancy, and breastfeeding.

**Keywords** breastfeeding; IGF-binding protein 4 and 5; insulin-like growth (IGF) factor signaling; involution
**Subject Category** Cancer

## Introduction

Pregnancy at a young age (before the age of 25) is associated with a reduction in the overall lifetime risk of developing breast cancer (Schedin, 2006). This observation represents the protective effect of pregnancy. However, pregnancy is also associated with a transient increase in risk of breast cancer in all women, which peaks at 6 years after pregnancy (Schedin, 2006). Further, the risk increases with the age of the mother at first pregnancy. Since women in developed countries tend to have children after the age of 30, pregnancy represents an important etiological factor in breast cancer today.

Further, a large analysis of the effect of breastfeeding using combined results from 47 studies, involving a total of 50,302 women, revealed that extended lactation is protective against breast cancer (Collaborative Group on Hormonal Factors in Breast, 2002). This study suggested that the cumulative risk of breast cancer could be reduced by half should the period of breastfeeding be increased (Collaborative Group on Hormonal Factors in Breast C, 2002). In addition, breastfeeding was found to be protective against more aggressive tumors (Faupel-Badger et al, 2013), but the mediators of the protective effect of lactation have not been identified.

The current definition of pregnancy-associated breast cancer (PABC) is empirically limited to breast cancers arising within 2 years of pregnancy. The 15-year survival of women diagnosed with breast cancer 1 and 2 years after giving birth is 38 and 51%, respectively, compared to 65% in age-matched nulliparous women (Schedin, 2006), suggesting the aggressive nature of the disease. The restricted time frame of diagnosis allowed by the current definition fuels the notion that PABC is a rare phenomenon. However, since the epidemiological data demonstrate that the increase in risk peaks 6 years following pregnancy, PABC arising beyond the 1- to 2-year window may represent the vast majority of cases. If so, PABC may represent a considerable fraction of breast cancers. In agreement with this notion, several studies indicate that the time frame after pregnancy for which a diagnosis of pregnancy-associated breast cancer is considered should be extended (Albrektsen et al, 2006; Andersson et al, 2009; Callihan et al, 2013).

One major breakthrough regarding PABC is the discovery that involution is tumorigenic (Lyons et al, 2011). Involution is a complex event where cell death of selected epithelial cells,

---

1 Division of Hematology/Oncology of the Icahn School of Medicine at Mount Sinai, Tisch Cancer Institute, New York, NY, USA
2 Department of Molecular Biology and Genetics, Aarhus University, Aarhus, Denmark
3 Department of Pathology of the Icahn School of Medicine at Mount Sinai, Tisch Cancer Institute, New York, NY, USA
4 Dubin Breast Center of the Icahn School of Medicine, Tisch Cancer Institute, New York, NY, USA
5 Department of Cell and Regenerative Biology, University of Wisconsin, Madison, WI, USA
6 Laboratory for Optical and Computational Instrumentation, University of Wisconsin, Madison, WI, USA
7 Department of Biostatistical Sciences, Icahn School of Medicine at Mount Sinai, New York, NY, USA
*Corresponding author. Tel: +1 212 241 9541; E-mail: doris.germain@mssm.edu

 

remodeling of the extracellular matrix, and adipogenesis must be coordinated. The signaling pathways involve a wide array of genes as well as the contribution of the innate immune system (Watson & Kreuzaler, 2011). Among the multiple factors implicated in involution, the increase in collagen has been linked to its pro-oncogenic potential (Lyons *et al*, 2011). However, post-involution mammary glands, also rich in collagen, are known to provide protection against tumor formation (Maller *et al*, 2013). One possibility to explain this dual effect of collagen is that collagen can adopt different architectures and that while some may be protective, others, such as tumor-associated collagen signature (TACS), promote tumor formation (Conklin *et al*, 2011). Therefore, these studies suggest that altered architecture of collagen contributes to the pro-tumorigenic effect in involution. Whether some oncogenes are activated specifically in the presence of collagen and/or constrained to involution remains unknown.

A key mediator of involution is the insulin-like growth factor-binding protein-5 (IGFBP-5) as demonstrated by the fact that involution is severely delayed in the IGFBP-5 knockout mice (Boutin-aud *et al*, 2004; Beattie *et al*, 2006; Ning *et al*, 2007; Akkiprik *et al*, 2008). IGFBP-5 is negatively regulated by the protease pappalysin-1 (PAPP-A) (Boldt *et al*, 2001; Laursen *et al*, 2001, 2007; Overgaard *et al*, 2001; Oxvig, 2015). A recent report showed that PAPP-A is overexpressed in all breast cancers (Mansfield *et al*, 2014), but whether it is an oncogene in the breast has not been tested. Further, the very high frequency of PAPP-A overexpression suggests that it may require particular conditions to become an oncogene. Considering that PAPP-A co-localizes with collagen (Chen *et al*, 2003) and cleaves IGFBP-5, we hypothesize that PAPP-A may contribute to the etiology of PABC at least in part by cleaving IGFBP-5 during involution.

While involution is reported to be involved in the progression of PABC, the potential contribution of gestation and lactation to the etiology of PABC has not been addressed. Since PAPP-A in addition to cleaving IGFBP-5 also cleaves IGFBP-4, which is expressed throughout all phases of pregnancy, its role during gestation and lactation should also be studied. Further, stanniocalcin-1 and stanniocalcin-2 (STC1 and STC2) have recently been shown to act as inhibitors of PAPP-A (Jepsen *et al*, 2015) and are found in breast-feeding milk (Tremblay *et al*, 2009). Therefore, we initiated this study to examine the effect of expression of PAPP-A in the mammary gland of virgin females and in parous females during gestation, lactation, and involution. Our findings indicate that PAPP-A is not oncogenic in virgin female mice but is a pregnancy-dependent oncogene. However, the oncogenic potential of PAPP-A is limited by lactation.

## Results

### Delayed mammary gland involution is observed in PAPP-A transgenic mice

PAPP-A is frequently overexpressed in breast cancer, but whether PAPP-A is an oncogene has never been determined. We first tested the expression of endogenous PAPP-A during the various phases of the mammary glands and found that it is expressed in virgin mice and during pregnancy but not during lactation or involution (Fig 1A). Therefore, to mimic its overexpression in breast cancer, we created transgenic mice overexpressing the PAPP-A gene in the mammary gland. To avoid unrelated effect due to insertion of the transgene, we initially analyzed three independent founder lines in parallel. Expressions of PAPP-A mRNA were found to be 20- to 30-fold higher than non-transgenic mice (Fig 1B), which is the average transgene expression using this promoter (Guy *et al*, 1992; Taneja *et al*, 2009). As predicted for the expression of a transgene driven by the MMTV promoter, the expression of PAPP-A was found to be similar throughout the phases of the mammary glands (Fig 1C). The expression of PAPP-A was confirmed by immunohistochemistry and, as expected for a secreted protease, found to localize to the lumen (Appendix Fig S1A). Since involution is implicated in PABC, and IGFBP-5 plays an important role during this phase of mammary gland development, we first focused our analysis on involution. While involution was nearly completed at day 12 in the non-transgenic females (Fig 1D and E, Appendix Fig S1B), we found the mammary glands of transgenic females at day 12 of involution resembled those of days 3–6 in non-transgenic glands (Fig 1D and E, Appendix Fig S1B). This delay in involution in transgenic females was also apparent at days 3 and 6 (Fig 1D and E, Appendix Fig S1B). PAPP-A expression was detected throughout involution time course (Fig 1F). Consistent with IGFBP-5 being a proteolytic target

**Figure 1.  Expression of PAPP-A in the mammary gland of transgenic mice leads to a delay in mammary gland involution.**

A     Endogenous mouse PAPP-A transcript levels in non-transgenic (NT) mouse mammary glands from virgin (V), pregnancy (P), lactation (L), involution (I) time point (*n* = 3 mice per group, triplicate experiments, mean ± SD). One-way ANOVA with Tukey's *post hoc* test: ****$P$ < 0.0001.

B     Human PAPP-A transcript levels in NT versus three lines of PAPP-A (PA) transgenic mice (*n* = 3 mice per group, triplicate experiments, mean ± SD). One-way ANOVA with Tukey's *post hoc* test: **$P$(PA1) = 0.0016, **$P$(PA4) = 0.0067, ***$P$(PA6) = 0.0009. All subsequent experiments were conducted in PA1 mice.

C     Transgenic human PAPP-A transcript level in PAPP-A transgenic mouse mammary glands from virgin (V), pregnancy (P), lactation (L), involution (I) time point (*n* = 3 mice per group, triplicate experiments, mean ± SD). One-way ANOVA with Tukey's *post hoc* test: all $P$ > 0.05.

D     Whole-mount analysis of involuting mammary glands (*n* = 12 mice; three mice per time point) at days 1, 3, 6, and 12 of involution in NT and PAPP-A females, scale bar: 1 mm.

E, F    (E) H&E sections and (F) immunohistochemistry for PAPP-A of involuting mammary gland time points, scale bar: 100 μm.

G     Immunoblot of mammary glands during involution in NT and PAPP-A females.

H     Quantification of IGFBP-5 from immunoblot shown in (G). Each bar represents the mean ± SD of three independent experiments. Unpaired *t*-test (two-tailed) between NT and PAPP-A at each time point: *$P$(day 12) = 0.0440.

I     Immunoblot and quantification of IGFBP-5 in mammary glands of 6-week-old virgin (*n* = 4 mice per group) NT versus PAPP-A females. Unpaired *t*-test (two-tailed): $P$ = 0.9720.

J     IGFBP-5 immunohistochemistry in mammary glands from 6-week-old virgin NT versus PAPP-A females, scale bar: 50 μm.

Source data are available online for this figure.

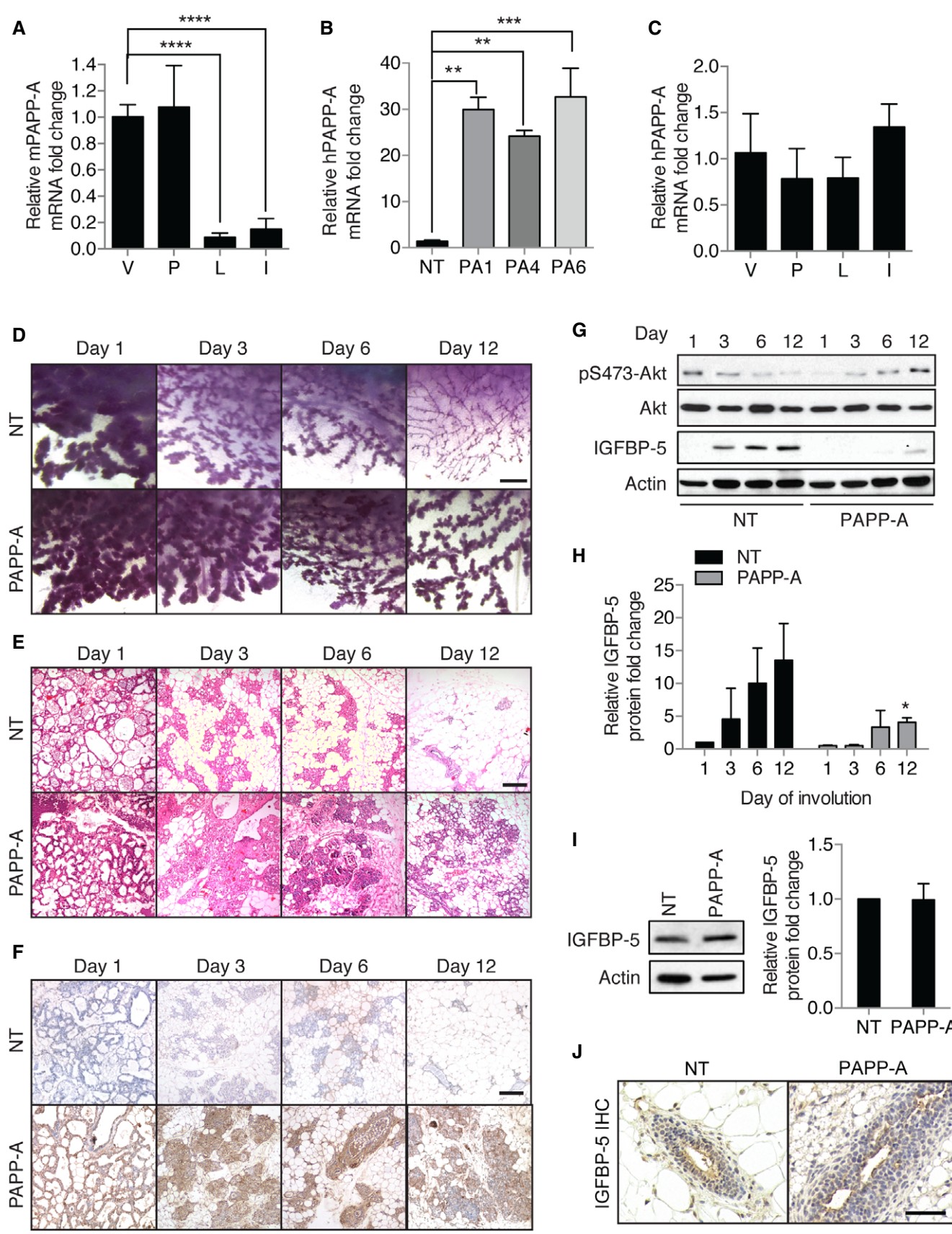

Figure 1.

of PAPP-A, un-cleaved IGFBP-5 was undetectable in the transgenic glands by immunoblotting at days 3 and 6 and only mildly detectable at day 12 (Fig 1G and H). Interestingly, in mammary glands of transgenic virgin females, despite the expression of PAPP-A, the level of un-cleaved IGFBP-5 was similar to that of non-transgenic mice (Fig 1I and J). This result suggests that involution provides a factor that promotes cleavage of IGFBP-5 by PAPP-A, which is not found in virgin glands. Since the delay in involution and the cleavage of IGFBP-5 specifically during involution were observed in all three transgenic founder lines, we concluded that the effect observed is due to the overexpression of PAPP-A and not to an indirect effect of the insertion of the transgene. Therefore, for all subsequent analyses we focused on the PA1 transgenic line, which expressed intermediate levels of PAPP-A compared to the other two lines (Fig 1B).

## The proteolytic activity of PAPP-A is increased by collagen deposition during mammary gland involution

IGF signaling is reported to increase the expression of collagen (Blackstock *et al*, 2014). As the ability of PAPP-A to cleave IGFBP-5 was observed in involuting, but not virgin glands, we reasoned that collagen may both affect IGBFP-5 proteolysis and promote the expression of collagen as a result of increased IGF signaling. To test this possibility, the collagen content was analyzed using Masson's trichrome stain, which revealed a substantial increase in the PAPP-A transgenic mice compared to non-transgenic females (Fig 2A and B and Appendix Fig S2A). In order to distinguish between preexisting and newly synthesized collagen, we performed RT–PCR and Picrosirius red staining, in which newly synthesized thin collagen fibers show a green birefringence under circularly polarized light. We found that compared to the non-transgenic females, the expression of collagen mRNA is significantly increased in the PAPP-A transgenic females during involution (Appendix Fig S3A). In agreement with the increased synthesis of collagen, we found an increase in green birefringence of Picrosirius red-stained sections in the transgenic females (Fig 2C and D, and Appendix Fig S2B) as well as an overall increase in collagen (Appendix Fig S3B and C), a result that is consistent with that obtained with Masson's trichrome staining (Fig 2A). The increase in collagen was confirmed using second harmonic generation (SHG) imaging, which relies on the autofluorescence of collagen (Fig 2E) (Conklin *et al*, 2011; Bredfeldt *et al*, 2014). The results were quantified by measuring the density of collagen around the ducts. We found a highly significant increase in the deposition of collagen surrounding the ducts in the PAPP-A transgenic females

compared to the non-transgenic females $(P = 6.013 \times 10^{-8})$ (Fig 2F). This result indicates that the expression of PAPP-A significantly increases the deposition of collagen during involution.

To investigate the effect of collagen on the ability of PAPP-A to cleave IGFBP-5, *in vitro* assays were performed using media from MCF-7 cells (control media) or MCF-7 cells stably expressing PAPP-A (PAPP-A media). The expression of PAPP-A transcript was determined by RT–PCR (Fig 3A) and protein levels in the culture supernatant by ELISA (Fig 3B) and Western blot (Fig 3C), confirming the secretion of PAPP-A. Incubation of recombinant IGFBP-5 (rIGBFP-5) with control medium (ctl) did not affect its level in either the presence or absence of collagen (Fig 3D, Appendix Fig S4A and B). In the absence of collagen, we found that the levels of rIGFBP-5 were reduced by only 20% after a 3-h incubation with PAPP-A (p-a) alone (Fig 3D, Appendix Fig S4A and B). However, upon co-incubation with collagen, rIGFBP5 levels decreased by 55% within the same time frame (Fig 3D, Appendix Fig S4A and B). As a control, the effect of another extracellular matrix protein, laminin was also tested and found to have no significant effect (Appendix Fig S4A and B). This result suggests that the presence of collagen increases rIGFBP-5 degradation by PAPP-A. In addition, the effect on endogenous IGFBP-5 in cells grown on plastic or on a surface coated with collagen was tested. Similarly, we found that the presence of collagen increased the degradation of IGFBP-5 in PAPP-A-expressing cells (Fig 3E).

The implication of this finding is that collagen may unleash the oncogenic potential of PAPP-A by increasing its proteolytic activity. To test this possibility, MCF-7 (ctl) and MCF-7^PAPP-A (p-a) breast cancer cells were added to a mixture of Matrigel and collagen at a ratio of 1:1 and injected into the fat pad of virgin nude mice. We found that while MCF-7 xenografts did not grow, cells expressing PAPP-A formed tumors (Fig 3F and Appendix Fig S5A). Further, we confirmed the level of IGFBP-5 to be reduced in these tumors (Fig 3G) and that the reduction in IGFBP-5 was associated with increased IGF signaling as measured by phosphorylated Akt (Fig 3G). To test the effect of collagen in a more physiological context, MCF-7 and MCF-7^PAPP-A cells were mixed in Matrigel and injected into the fat pad of virgin mice, where the concentration of endogenous collagen is minimal. This analysis showed that the size of palpable tumors was similar in both groups (Fig 3H and Appendix Fig S5B). When the same cells were injected into the fat pad of involuting mammary glands, we found that the PAPP-A-expressing cells grew significantly faster $(P = 0.015)$ (Fig 3I and Appendix Fig S5C). While recruitment of macrophages and immune cells in the fat pad during involution is likely to contribute to the pro-tumorigenic effect of involution, since such recruitment is

---

**Figure 2.  Increased collagen deposition is observed during mammary gland involution in PAPP-A transgenic mice.**

A   Masson's trichrome stain on involuting mammary glands from NT and PAPP-A mice. Blue: collagen. Scale bar: 100 μm.

B   Quantification of collagen by Masson's trichrome stain from (B) (*n* = 3 mice, each point represents the average of three determinations per mouse per time point). Each bar represents the mean ± SD. Unpaired *t*-test (two-tailed): *$P$(day 6) = 0.0408, *$P$(day 12) = 0.0440.

C   Newly synthesized collagen (thin fibers' green birefringence) by circularly polarized light microscopy on Picrosirius red-stained slides of involuting mammary glands from NT and PAPP-A mice, scale bar: 100 μm.

D   Quantification of newly synthesized collagen fibers by green birefringence from (D) (*n* = 3 mice, each point represents the average of three determinations per mouse per time point). Each bar represents the mean ± SD. Unpaired *t*-test (two-tailed): **$P$(day 6) = 0.0095, *$P$(day 12) = 0.0164.

E   Second harmonic generation (SHG) imaging of collagen on magnified ducts of histological sections shown in (A). Region of interest (ROI) of ducts acquired as negative images of nonspecific auto-fluorescence, scale bar: 50 μm.

F   Graph of collagen intensity relative to the distance from duct edge of NT and PAPP-A transgenic involuted mammary ducts as imaged by SHG in (E) at day 12 (*n* = 3 mice, 18 ducts total). Each bar represents the mean ± SD. Unpaired *t*-test (two-tailed) for the two groups: ****$P$ = 6.013 × 10^{-8}.

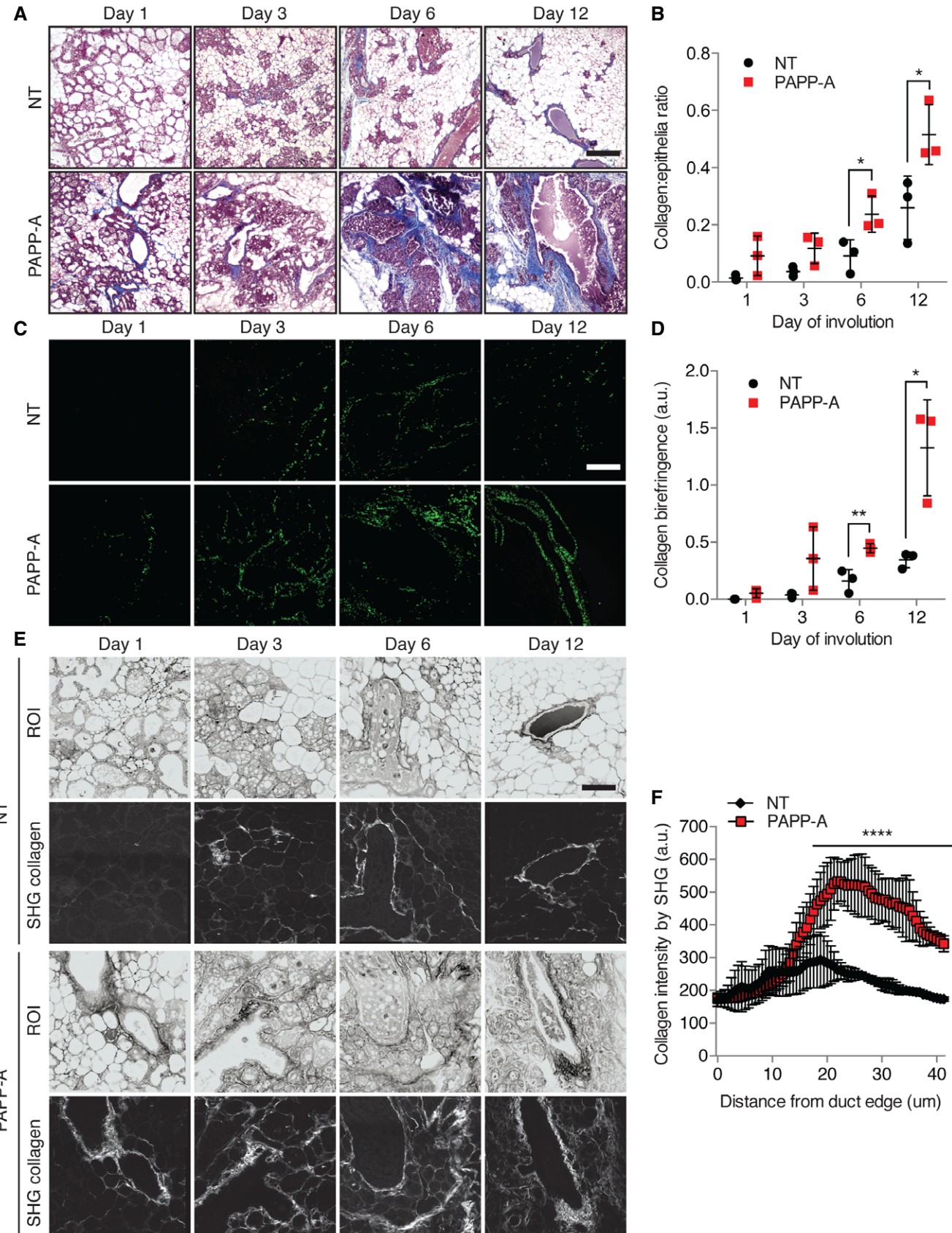

Figure 2.

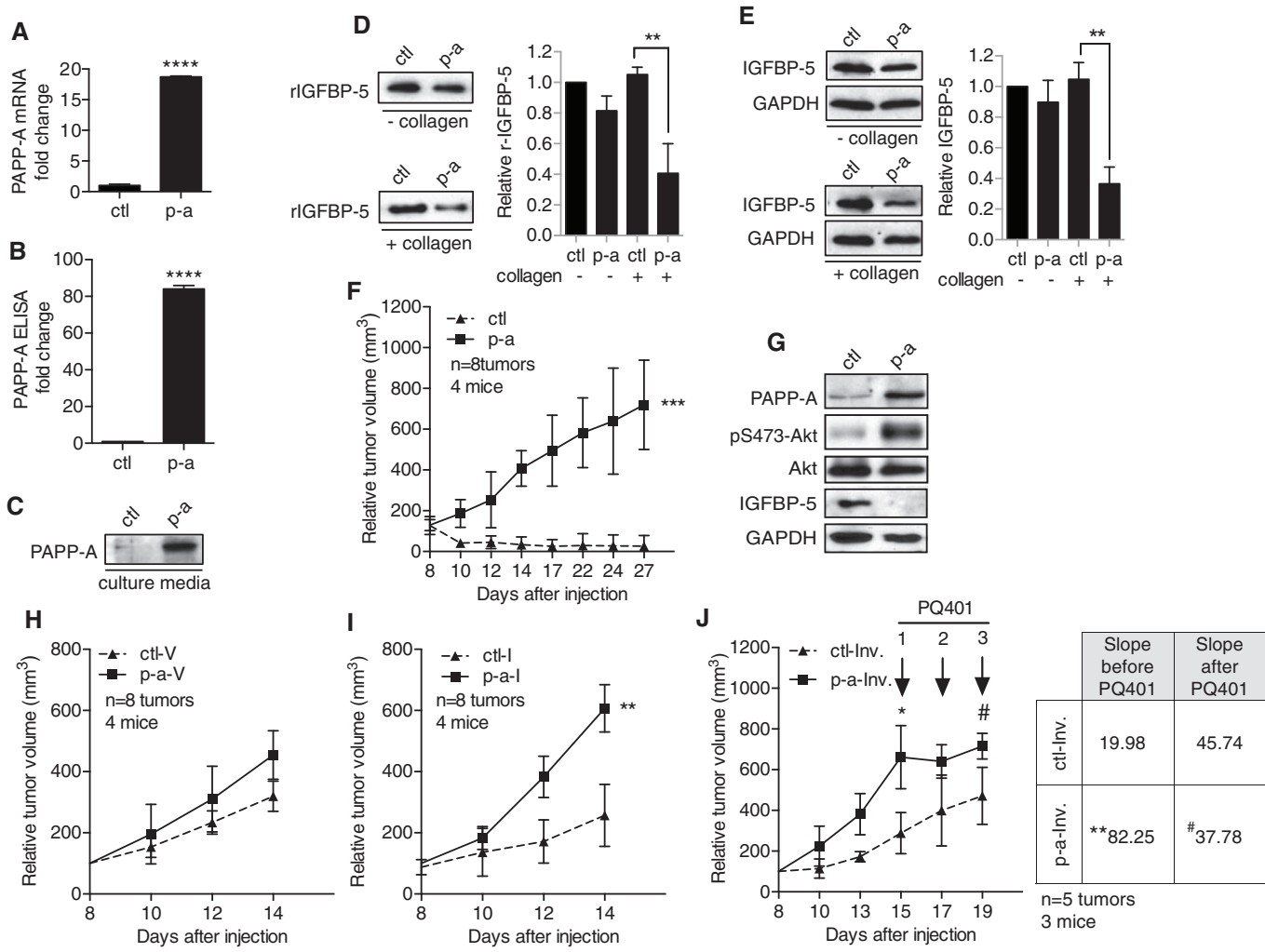

**Figure 3. Collagen enhances the proteolytic activity of PAPP-A *in vitro* and *in vivo* tumor growth.**

A   PAPP-A transcript levels in MCF-7 control cells (ctl) and MCF-7 cells stably expressing PAPP-A (p-a) (*n* = 3 independent experiments, mean ± SD). Unpaired *t*-test (two-tailed): ****P* < 0.0001.

B   Levels of secreted PAPP-A in cell media as detected by ELISA (*n* = 3 independent experiments, mean ± SD). Unpaired *t*-test (two-tailed): ****P* < 0.0001.

C   Immunoblot of PAPP-A in culture media from the cells as in (A) and (B).

D   Immunoblot and quantification of rIGFBP-5 following an incubation of 3 h in culture media from MCF7 cells (ctl or p-a) in the absence or presence of collagen (*n* = 3 independent experiments, mean ± SD). Unpaired *t*-test (two-tailed): ***P* = 0.0050.

E   Immunoblot and quantification of endogenous IGFBP-5 levels of MCF7 ctl or p-a cells, in the absence or presence of collagen (*n* = 3 independent experiments, mean ± SD). Unpaired *t*-test (two-tailed): ***P* = 0.0120.

F   Tumor volumes of ctl and p-a xenografts mixed with Matrigel and collagen (*n* = 4 mice, eight tumors total), ****P* = 0.0008. For analysis details see Materials and Methods section.

G   Immunoblot of representative xenografts at day 27 from (F).

H, I   (H) Relative tumor volume of ctl and p-a xenografts injected in fat pad of virgin female mice (-V) (*n* = 4 mice, eight tumors total), *P* = 0.0811, or (I) injected in actively involuting fat pad (-Inv.) (*n* = 4 mice, eight tumors total), ***P* = 0.0092.

J   Relative tumor volume of ctl and p-a xenografts injected in actively involuting fat pad treated with three treatments of PQ401, starting at day 15 and ending at day 19 (*n* = 3 mice, five tumors total), **P* = 0.0031, #*P* = 0.4270.

Data information: Statistical test: for (F, H and I), each point represents the mean ± SD. Unpaired *t*-test (two-tailed) was applied to calculate statistical significance at the end point or (J) at the slope (tumor growth rate) before or after PQ401 administration.

Source data are available online for this figure.

present in both MCF-7 and MCF-7^PAPP-A xenografts, our result supports the conclusion that the expression of PAPP-A confers an additional oncogenic effect. Finally, since cleavage of IGFBP-5 by PAPP-A activates IGF signaling, we tested the effect of the anti-IGF receptor drug PQ401 on the growth of xenografts in involuting glands injected with either MCF-7 or MCF-7^PAPP-A cells. We found that anti-IGF therapy had no significant effect on the growth rate of MCF-7 xenografts before and after treatment (*P* = 0.01). In contrast, anti-IGF therapy significantly reduced the growth rate of xenografts expressing PAPP-A (*P* = 0.0001) (Fig 3J and Appendix Fig S5D).

Collectively, these results suggest that increased collagen deposition during involution facilitates the degradation of IGFBP-5 by PAPP-A. Since elevated IGF signaling further increases the deposition of collagen, we propose that the abnormal expression of PAPP-A during involution establishes a pro-tumorigenic positive feedback loop between IGF signaling and collagen.

### Low IGFBP-5 levels and a TACS-3 signature are hallmarks of PABC in PAPP-A transgenic mice

The effect of collagen on the proteolytic activity of PAPP-A raises the possibility that PAPP-A acts as a pregnancy-dependent oncogene. We therefore compared the frequency of mammary tumors in age-matched PAPP-A transgenic virgins and their parous counterparts. Hyper-proliferative lesions were detected exclusively in the mammary glands of the parous groups, while virgin mammary glands contained no detectable lesions (Fig 4A and B). Palpable mammary tumors were observed in female transgenic mice starting at 4 months of age, with an average of 8.4 months (Fig 4C and D), and were confirmed to have low expression of IGFBP-5 and increased phosphorylated Akt (Fig 4D). Further, tumors were observed in transgenic females shortly (3–7 weeks) after their last pregnancy. The PAPP-A mammary tumors were found to be adenocarcinomas with metaplastic features. One trivial explanation for

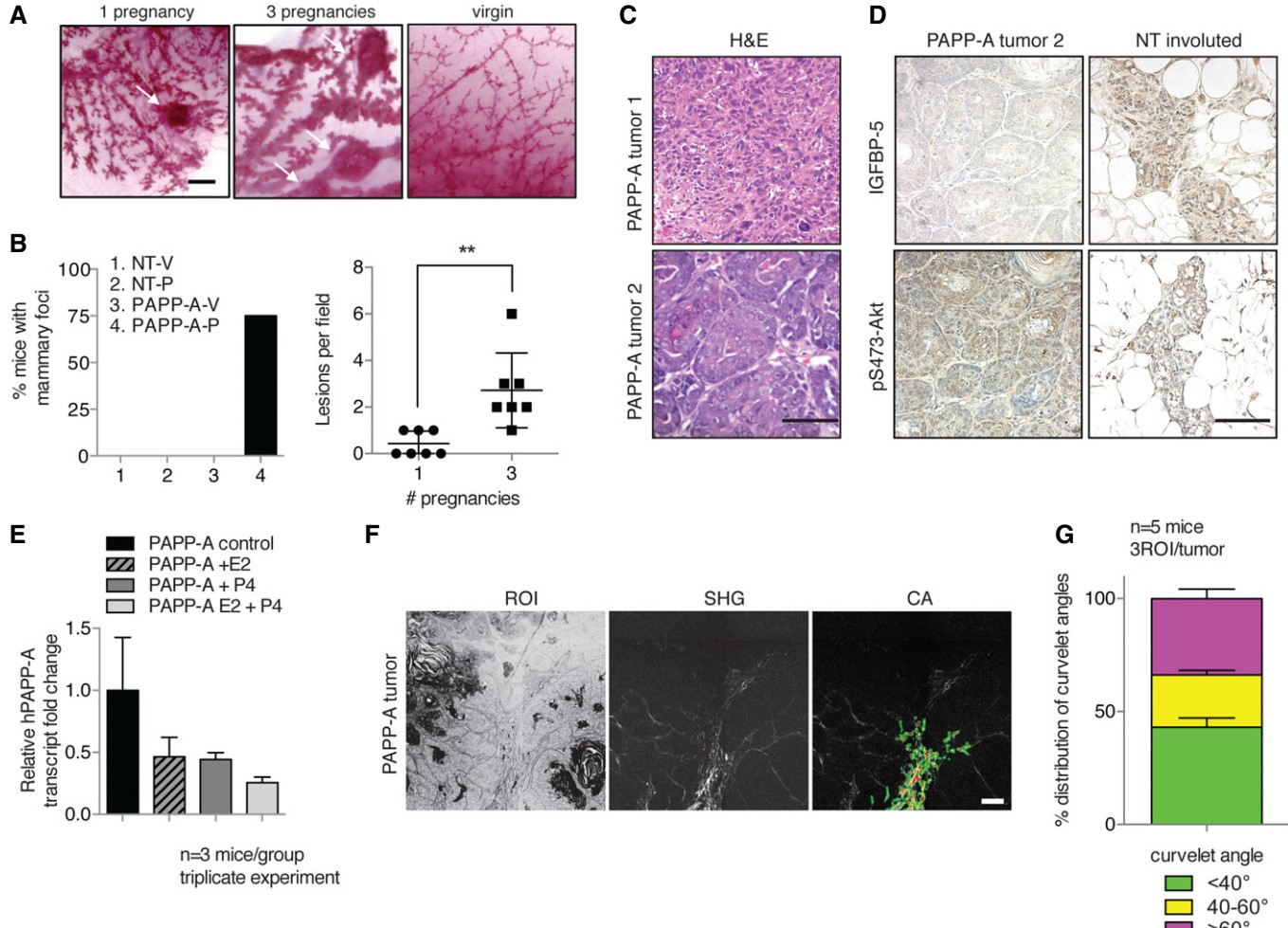

**Figure 4. Pregnancy-associated PAPP-A mammary tumors are characterized by low IGFBP-5 levels and collagen reorientation.**

A  Whole-mount sections of PAPP-A mammary glands after 1 or 3 pregnancies and PAPP-A age-matched virgin. White arrows indicate lesions. Scale bar: 1 mm.

B  Left panel: Frequency of mammary lesions in NT virgin (-V), NT pregnancy (-P) and PAPP-A virgin (-V), PAPP-A pregnancy (-P) glands (n = 8 mice except for PAPP-A-P: n = 14 mice, seven mice with one pregnancy, seven mice with three pregnancies). Right panel: Number of lesions in PAPP-A-P group subdivided into 1 and 3 pregnancies (n = 7 mice, mean ± SEM). Unpaired *t*-test (two-tailed): **P = 0.0038.

C  Histological sections of two different histologies of a late-stage (~14 months) postpartum PAPP-A mammary tumor. Scale bar: 100 μm.

D  IGFBP-5 and pS473-Akt immunohistochemistry of tumor 2 from (C) and a NT involuted gland, scale bar: 100 μm.

E  PAPP-A transgenic mouse mammary glands' PAPP-A transcript levels in untreated control, treated with estrogen (E2), progesterone (P4), or with both E2 and P4 for 1 week. (n = 3 mice, triplicate experiment, mean ± SD). One-way ANOVA with Tukey's *post hoc* test: all P > 0.05.

F  Imaging of collagen by SHG of PAPP-A mammary tumor (n = 5 tumors), scale bar: 200 μm.

G  Analysis of curvelet angle distribution for collagen alignment (n = 5 tumors from five mice, 3ROI analyzed per tumor, mean ± SEM).

this observation is that since the MMTV promoter is inducible by hormones (Ponta *et al*, 1985), the increase in tumors following pregnancy simply reflects an increased expression of the transgene. However, the hormone-sensitive region of the MMTV promoter used to establish our PAPP-A transgenic mice has been deleted (Ponta *et al*, 1985). In agreement with the hormonal insensitivity of our model, treatment with neither estrogen or progesterone nor both increased PAPP-A levels (Fig 4E). In fact, we found that PAPP-A levels were lower in virgin PAPP-A transgenic mammary glands in the presence of either estrogen, progesterone, or both (Fig 4E). These results therefore rule out an indirect effect of transgene expression levels on the incidence of mammary tumors following pregnancy.

Having established that the mammary glands of PAPP-A transgenic females show an increased deposition of collagen during involution, we next performed analyses for tumor-associated collagen signatures (TACS) (Provenzano *et al*, 2006). Importantly, TACS measures not only the density of collagen but also the orientation of collagen fibers relative to the tumor border. TACS are classified as TACS-1, TACS-2, or TACS-3; the radially oriented collagen fibers characterize the TACS-3 signature, which is strongly associated with more aggressive breast cancers (Conklin *et al*, 2011). Using the CurveAlign software developed to identify TACS-3 regions (a collagen fiber "curvelet" angle of > 60 degrees), we found TACS-3 regions in 40% of the cell-matrix border of the PAPP-A mammary tumors (Fig 4F and G). This finding suggests that the deposition of collagen induced by PAPP-A favors the formation of TACS-3.

### Gestation and multiple pregnancies amplify the oncogenic effect of PAPP-A

Our data indicate that the formation of spontaneous mammary tumors in PAPP-A transgenic mice is driven by the proteolytic degradation of IGFBP-5 during involution. The initial focus on involution arose from the reported role of involution in PABC (Lyons *et al*, 2011) and the importance of IGFBP-5 in this phase of pregnancy. However, a contribution of the gestation phase of pregnancy cannot be ruled out. Since IGFBP-4 is a major substrate of PAPP-A and is expressed in all phases of mammary gland development, we tested the effect of PAPP-A on the proliferation of mammary glands during gestation. First, we confirmed that IGFBP-4 and IGFBP-5 are degraded by PAPP-A during gestation (Fig 5A) and that this is associated with increased phosphorylation of Akt as

detected by Western blot and immunohistochemistry (Fig 5A and B). As we observed during involution, collagen deposition is also accelerated in the PAPPA transgenic mice during this phase of pregnancy (Fig 5C and D, and Appendix Fig S2C). However, this analysis also revealed that the expression of PAPP-A had no significant effect on ductal morphology compared to non-transgenic females that undergo their first round of gestation (Fig 5E). No morphological changes were observed despite the fact that between days 3 and 9 of gestation, a significant activation of the proliferative signaling pathway as measured by phosphorylation of STAT-5a/b and Akt was observed in the PAPP-A transgenic mice compared to non-transgenic mice (Fig 5F). Noticeably, an abrupt reduction in these proliferative signals was observed on day 12 in both transgenic and non-transgenic females (Fig 5F), which was also observed by IHC (Fig 5B). This observation suggests that the ability of PAPP-A to increase these pathways is transient.

We next tested the effect of multiple rounds of pregnancy. A more pronounced proliferation of the ductal tree was apparent in transgenic females during their third pregnancy (Fig 5G). Further, the levels of phosphorylated STAT5 and Akt were also further amplified (Fig 5H). Hence, these results suggest that the effect of PAPP-A during gestation is cumulative over several pregnancies and may, in addition to the contribution by involution, also contribute to its oncogenic potential. However, an inhibitory event occurring late during gestation limits the contribution of this phase of pregnancy.

### Lactation suppresses tumor growth and PAPP-A activity

In order to complete our analysis of the effect of PAPP-A during pregnancy, we next focused on lactation. Tumor incidence in PAPP-A transgenic mice was evaluated relative to the length of lactation: long lactation (more than 3 weeks, range = 21–24 days, average = 21.25 days) and short lactation (< 2 weeks, range = 0–13 days, average = 1.076 days). Strikingly, none of the mothers that nursed their pups for an extended period of time developed tumors, while 43% of those that either had not nursed their pups or lactated for a short period developed mammary tumors (Fig 6A). Importantly, females that had multiple pregnancies with long lactation, followed by a pregnancy without lactation, still develop mammary tumors. This result suggests that the cumulative effect of multiple gestations on the proliferation of the mammary ductal tree (Fig 5G) contributes to the formation of mammary tumors. Furthermore, this unexpected finding raised the possibility that the activity of PAPP-A is

**Figure 5.   Transient activation of pregnancy signaling pathways in PAPP-A transgenic mammary glands may dictate tumor incidence.**

A   Western blot analysis of NT and PAPP-A mammary glands at day 1 of pregnancy.

B   pSer473-Akt immunohistochemistry on mammary glands at days 3, 9, and 12 of pregnancy of NT and PAPP-A transgenic mice (*n* = 9 mice; three mice per time point), scale bar: 25 μm.

C   Masson's trichrome stain (MTS) of mammary glands at indicated time points of pregnancy of NT and PAPP-A transgenic mice, blue: collagen, scale bar: 100 μm.

D   Quantification of collagen by MTS from (C) (*n* = 3 mice, average of three determinations per mouse per time point). Each bar represents the mean ± SD. Unpaired *t*-test (two-tailed): ****P(day 9) < 0.001, **P(day 12) = 0.0032.

E   Whole-mount analyses of mammary glands at days 3, 9, and 12 of first pregnancy in NT and PAPP-A females, scale bar: 0.5 mm.

F   Immunoblot of mammary glands at indicated time points of first pregnancy.

G   Representative whole-mount analyses of mammary glands at day 9 of first (P1) or third (P3) pregnancy from PAPP-A females (*n* = 6 mice; three mice per time point), scale bar: 0.5 mm.

H   Immunoblot analyses of mammary glands at day 9 of first (P1) or third (P3) pregnancy.

Source data are available online for this figure.

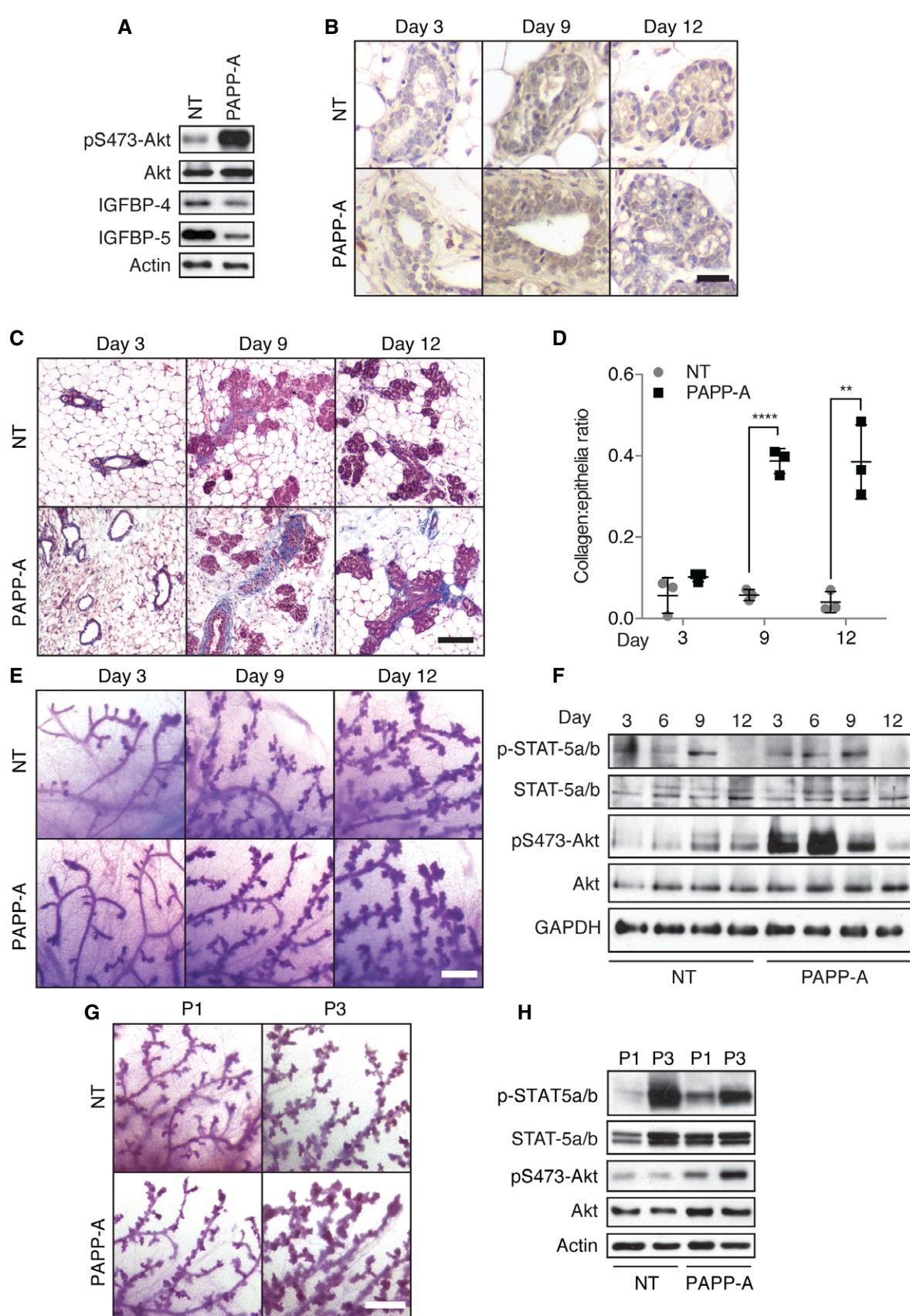

**Figure 5.**

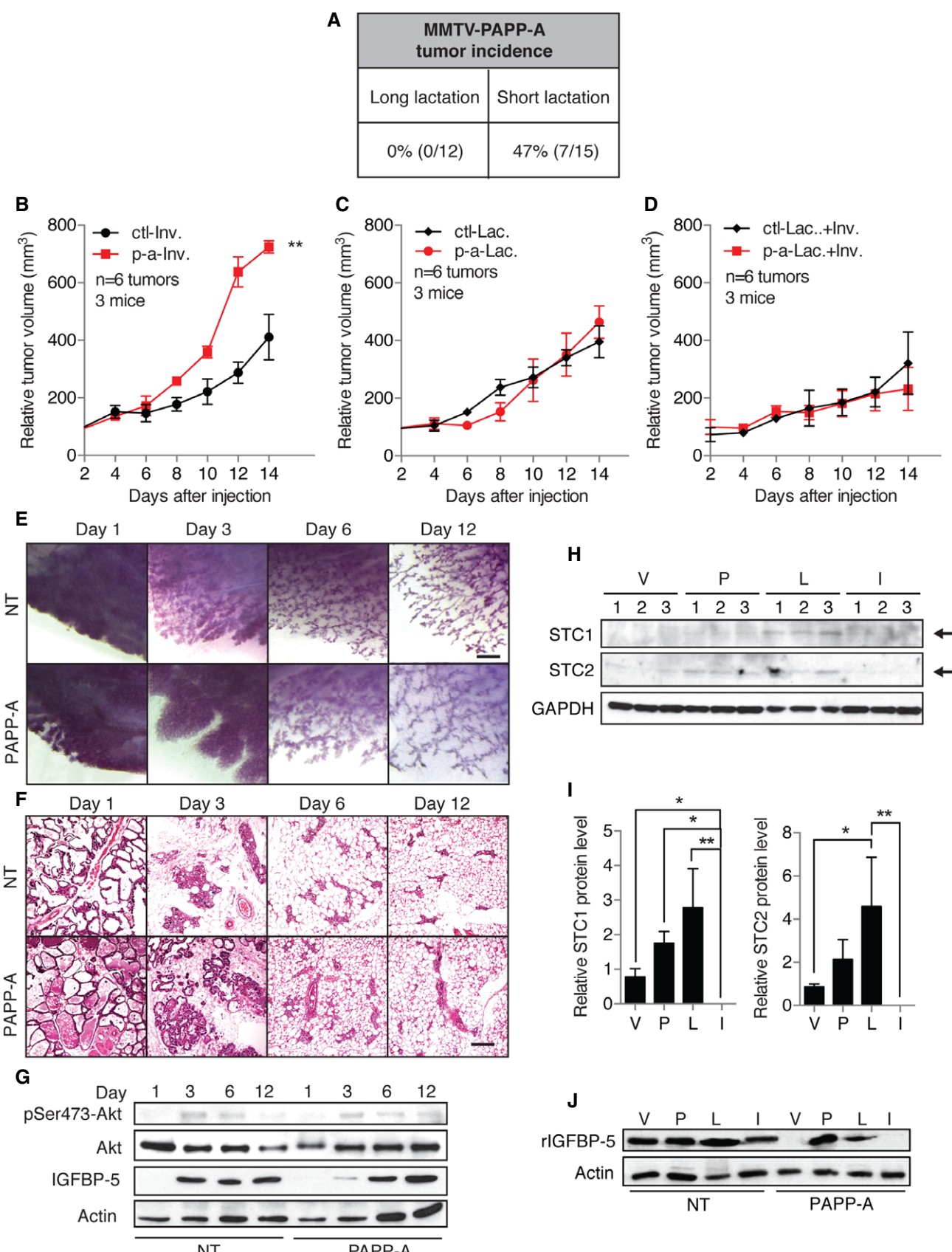

Figure 6.

**Figure 6. Lactation inhibits the oncogenic effect of PAPP-A.**

A    Incidence of spontaneous mammary tumor formation in PAPP-A females relative to the duration of lactation. Long lactation of > 3 weeks ranges from 21 to 24 days, with an average of 21.25 days, while short lactation of < 2 weeks ranges from 0 to 13 days, with an average of 1.076 days. Fisher's exact test: $P = 0.0170$.

B–D    Growth rate of ctl and p-a xenografts injected in fat pad of actively involuting mammary glands without any prior lactation (-Inv.), of actively lactating (-Lac.), and of actively involuting mammary glands following 2 weeks of prior lactation (-Lac.+Inv.), ($n = 3$ mice, 6 tumors total, mean ± SD). Unpaired $t$-test (two-tailed) was applied to calculate statistical significance at the end point. **$P = 0.0027$ for (B), $P = 0.4359$ for (C), and $P = 0.3066$ for (D).

E    Whole-mount analyses of involuting mammary glands ($n = 12$ mice; three mice per time point) at days 1, 3, 6, and 12 of involution in NT and PAPP-A females, scale bar: 1 mm.

F    H&E sections of corresponding mammary glands from (E), scale bar: 100 μm.

G    Immunoblot of mammary glands during involution in NT and PAPP-A females.

H    Immunoblot for STC1 and STC2 from mammary gland extracts of virgin (V), at pregnancy (P), at lactation (L), or at involution (I) performed in triplicate mice per group.

I    Quantification of immunoblot for STC1 or STC2 from (H). Error bars are mean ± SD. Left panel (STC1) *$P = 0.0148$ (V versus I), *$P = 0.0293$ (P versus I), **$P = 0.0021$ (L versus I). Right panel (STC2) *$P = 0.0248$ (V versus L), **$P = 0.0079$ (L versus I).

J    Immunoblot of rIGFBP-5 following proteolysis assay using NT or PAPP-A extracts from V, P, L or I mammary glands.

Source data are available online for this figure.

suppressed during lactation and that the degree of its suppression is directly linked to the length of lactation.

To better understand the response of breast cancer cells to the pregnant gland niche, we injected MCF-7 (ctl) or MCF-7$^{PAPP-A}$ (p-a) into the mammary fat pads of nude mice in three different groups: (i) involuting glands with no prior lactation (Fig 6B), (ii) actively lactating glands (Fig 6C), and (iii) involuting glands that underwent 2 weeks of lactation prior to the initiation of involution (Fig 6D). Consistent with the previous finding (Fig 3I), compared to control cells, cells expressing PAPP-A grew significantly faster in involuting glands with no prior lactation (Fig 6B and Appendix Fig S6A). However, the difference in growth rate between the two cell lines was abolished when injected into actively lactating glands or involuting glands that had undergone 2 weeks of lactation prior to the initiation of involution (Fig 6C and D, Appendix Fig S6B and C).

These results indicate that lactation may abolish the effect of PAPP-A during involution. Since the delay in involution, which we observed previously (Fig 1D and E), was in females that had lactated for only 2 days, we repeated the same time course of involution with the exception that PAPP-A females lactated for 2 weeks prior to the initiation of involution. Strikingly, following a long lactation, the delay in involution was fully abolished (Fig 6E and F, and Appendix Fig S1C). Further, the degradation of IGFBP-5 was also inhibited (Fig 6G). These observations demonstrate the suppressive effect of lactation on morphological changes instigated by the expression of PAPP-A and strongly support the presence of an inhibitor of PAPP-A during lactation.

Recently, the glycoprotein stanniocalcin-2 (STC2) was reported to act as a potent inhibitor of PAPP-A and prevent the ability of PAPP-A to cleave IGFBP (Jepsen *et al*, 2015). STC2 is associated with a reduction in the viability of human breast cancer cells (Raulic *et al*, 2008). Moreover, expression of STC2 in human breast cancer correlates with good outcome (Bouras *et al*, 2002). Interestingly, STC1 has been also found to be an inhibitor of PAPP-A (Kloverpris *et al*, 2015). The expression of STC1 is lost in breast cancer (Welcsh *et al*, 2002). Further, STC1 is produced by the ovaries and can be detected at high levels in the serum only during pregnancy and lactation, but not during involution (Deol *et al*, 2000). However, the presence of STC2 and/or STC1 in mammary gland at specific stages of pregnancy has never been determined.

We therefore examined the levels of STC1 and STC2 transcripts and proteins in the mammary gland of virgin, pregnant, lactating or involuting wild-type females. We found no stage-associated change

in STC1 or STC2 mRNA levels (Appendix Fig S7A). However, at the protein level, compared to virgin or involuting glands, both STC1 and STC2 were elevated during late pregnancy and lactation (Fig 6H and I).

We next tested the proteolytic activity of PAPP-A present in extracts from PAPP-A transgenic mammary gland from virgin, pregnant, lactating, and involuting glands, by measuring degradation of recombinant IGFBP-5 (rIGFBP-5). Longer (18 h) incubation time was required to detect PAPP-A activity in tissue extracts than in cells lines (3 h, Fig 3). While using such long incubation, the extracts from virgin glands, which express low levels of STC1 and STC2, degraded rIGFBP-5, while extracts from pregnant or lactating glands did not (Fig 6J). We next immunodepleted STC1 and/or STC2 from the pregnant and lactating mammary extracts of the PAPP-A transgenic mice to test whether removal of the inhibitors restores the ability of PAPP-A to cleave rIGFBP-5. We found however that rather than increasing rIGFBP-5 degradation, we found its stabilization (Appendix Fig S7B). We reasoned that since STC2 and STC1 bind to PAPP-A, the immunoprecipitation of either STC may simultaneously deplete PAPP-A from the extract. Indeed, as determined by ELISA, immune depletion of STC also depleted PAPP-A (Appendix Fig S7C).

Taken together, these results indicate that expression of STC1 and STC2 during late pregnancy and lactation correlates with reduced PAPP-A activity, which are therefore prime candidates for the observed inhibition of its oncogenic potential during these phases.

**Endogenous PAPP-A has limited effect on the regulation of IGFBP-5 during mammary gland development**

While our model mimics the overexpression of PAPP-A in breast cancer, its endogenous level during the different phases of the mammary gland has never been described. We therefore determined the levels of PAPP-A in virgin, pregnant, lactating, and involuting mammary glands. We found that endogenous PAPP-A levels are higher in virgin and pregnant mice but were reduced during lactation and involution (Fig 1A). To determine whether PAPP-A levels inversely correlate with the expression of IGFBP-5, we performed a Western blot of endogenous IGFBP-5. In agreement with our finding in the MMTV-PAPP-A transgenic mice (Fig 1I), the levels of IGFBP-5 are higher in virgin despite the expression of endogenous PAPP-A (Appendix Fig S7D). Levels of IGFBP-5 protein were slightly lower in pregnancy than in virgin gland most likely due to some

deposition of collagen during pregnancy. However, endogenous IGFBP-5 levels were lowest during lactation (Appendix Fig S7D) despite the lack of expression of PAPP-A during this phase (Fig 1A). This result is consistent with the finding that IGFBP-5 transcription is inhibited during lactation (Boutinaud *et al*, 2004). Further, IGFBP-5 levels raised in involution, which also confirms the reported increase in the transcription of IGFBP-5 during involution. This result indicates that in a normal mammary gland, the level of IGFBP-5 rise during involution and is unopposed by PAPP-A. Therefore, the abnormal presence of PAPP-A during involution and its ability to degrade IGFBP-5 represents a gain of function.

**Postpartum breast cancers display a PAPP-A/TACS-3 signature and decreased IGFBP-5 levels**

Collectively, the analysis of the effect of expression of PAPP-A both *in vitro* and *in vivo* suggests that PAPP-A is a pregnancy-dependent oncogene that promotes the formation of mammary tumors characterized by a TACS-3 signature and low IGFBP-5 levels. To determine whether these observations can be validated in human PABC, we performed a sub-analysis of PAPP-A levels by immuno-histochemistry specifically in premenopausal patients and according to their history of pregnancy. In addition, we also stained for IGFBP-5 since our data in mice indicate that positivity for PAPP-A does not necessarily correlate with decrease in IGFBP-5 as collagen is needed for PAPP-A to cleave IGFBP-5.

Representative images of the same tumors stained for PAPP-A and IGFBP-5 are shown in Fig 7A. Patients were divided into two groups, those who had at least one full-term pregnancy (parous) and those who never had children (nulliparous). Baseline analysis of the two groups revealed no statistical difference in the distribution of age at diagnosis, (Fig 7B), tumor volume, or stage (Appendix Fig S8A). Of the 28 patients in the parous group, the information on the date of the last pregnancy was available for 21

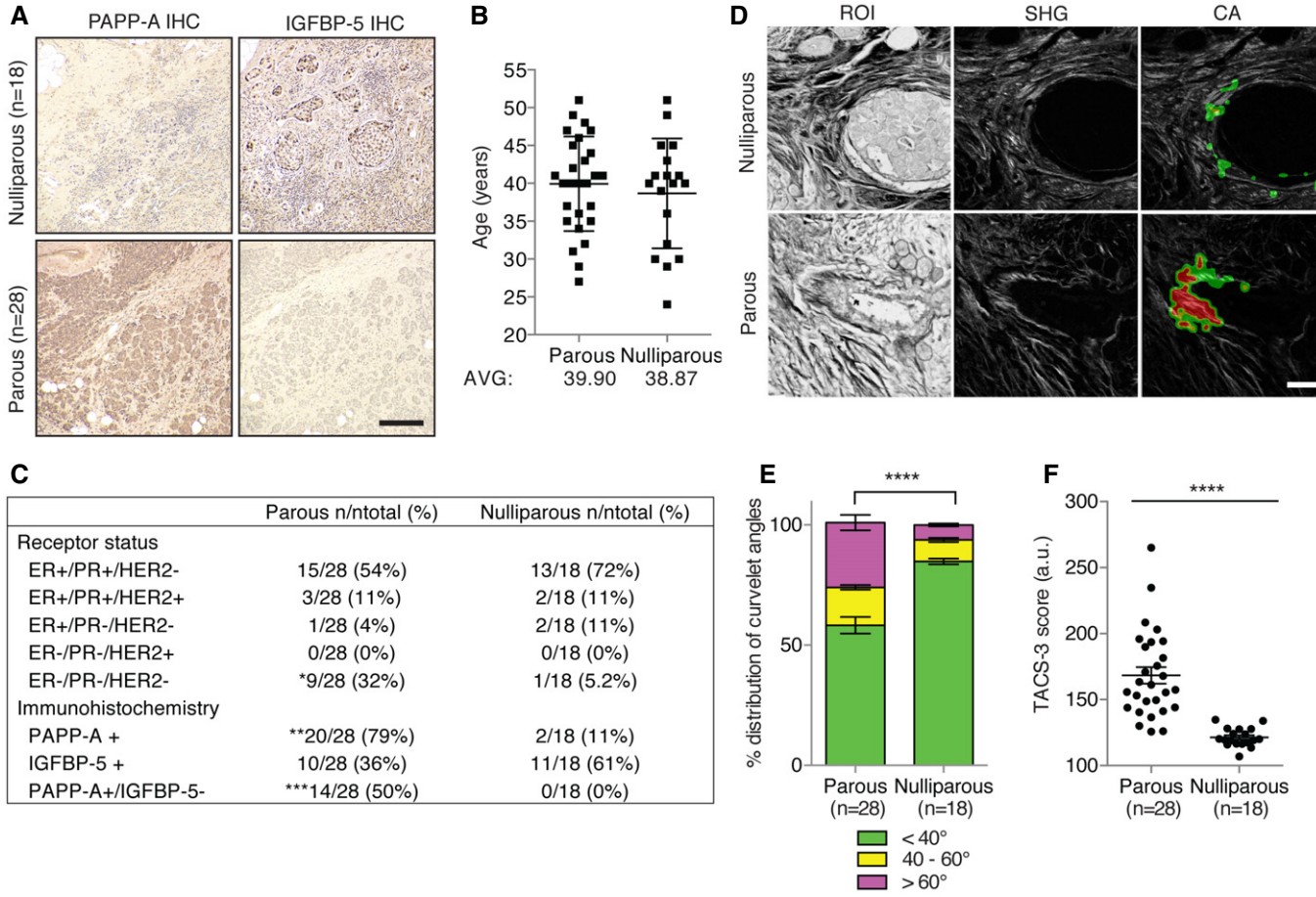

**Figure 7. A strong TACS-3 signature and decreased IGFBP-5 levels characterize postpartum PAPP-A positive breast cancers.**

A  Representative PAPP-A and IGFBP-5 immunohistochemistry of nulliparous (*n* = 18) and parous (*n* = 28) breast cancer patients. Scale bar: 200 μm.
B  Age range of breast cancer patients in parous and nulliparous groups available for analysis. Error bars represent mean ± SD.
C  Receptor status and frequency of positive/negative immunostaining in patient slides positive for PAPP-A, IGFBP-5, and PAPP-A positive/IGFBP-5 negative. Two-way ANOVA with Bonferroni's test: *$P = 0.047$, **$P = 0.0001$, ***$P = 0.0005$.
D  SHG image of collagen from patient slides, scale bar: 50 μm.
E  Distribution of collagen curvelet angles, as calculated by CurveAlign software, of breast cancer sections from parous and nulliparous patients. One-way ANOVA with Tukey's *post hoc* test: ****$P < 0.0001$ for < 40°, $P = 0.379$ for 40° − 60°, and ****$P < 0.0001$ for > 60°.
F  TACS-3 signature score of breast cancer from parous and nulliparous patients. Unpaired *t*-test (two-tailed): ****$P < 0.0001$.

patients and revealed an average time between the last pregnancy and breast cancer diagnosis of 6.2 years (range = 0.25–22 years). In term of sub-type as defined by receptor status (ER, PR, and Her2), we found a significant increase in triple-negative breast cancers in the parous group (32%) compared to the never pregnant group (5.2%) (Fig 7C). However, information on the length of breastfeeding was not available for any patients in the parous group.

This analysis showed that the expression of PAPP-A was observed in 79% of patients who had children but only in 11% of nulliparous patients (Fig 7C). The level of IGFBP-5 was also determined in this cohort of patients. We found an inverse correlation between PAPP-A levels and IGFBP-5 where no patient in the nulliparous group had tumor that had both high PAPP-A and low IGFBP-5, while 50% of patients who had children did (Fig 7C). The analysis of this patient population was extended to the analysis of TACS-3 by SHG microscopy. We found a highly statistically significant higher rate of TACS-3 in the parous group relative to the nulliparous group ($P = 0.0001$) (Fig 7D–F and Appendix Fig S8B).

Collectively, these results show a strong correlation between history of pregnancy and breast cancers that are characterized by a PAPP-A/TACS-3/IGFBP-5 signature. We propose that this signature combined with the effect of PAPP-A during involution may identify PABC.

## Discussion

The existence of PABC as a distinct breast cancer sub-type remains controversial as it implies that some oncogenes can only unleash their proliferative potential in the context of a breast that had undergone gestation, lactation, and/or involution. Data presented here support the notion that PAPP-A is a pregnancy-dependent oncogene.

A model of the regulation of endogenous PAPP-A in the mammary gland and the effect of its overexpression in PABC is shown in Fig 8. In non-transgenic mice (Fig 8A), PAPP-A is expressed at low levels during pregnancy but is absent during

**Figure 8. Schematic diagram of a mechanism by which PAPP-A may act as a pregnancy-dependent oncogene.**

A, B   The diagram represents PAPP-A and IGF signaling over the course of pregnancy in a (A) normal mammary gland and (B) when PAPP-A is overexpressed. See text for details.

involution. This basal expression of PAPP-A may contribute to the activation of IGF signaling early during gestation, although the low level of collagen during this period limits the activity of PAPP-A. Further, since STC1 and STC2 have been identified as inhibitors of PAPP-A and that they are produced during late pregnancy and lactation, it is tempting to speculate that the expression of STC 1 and STC2 late during gestation and lactation is able to fully inhibit the basal activity of PAPP-A. As a result, in the absence of either preexisting active PAPP-A produced during pregnancy or newly synthesized PAPP-A during involution, the rise in IGFBP-5 mRNA levels during involution is unopposed by proteolysis allowing inhibition of IGF signaling and involution (Fig 8A). In the MMTV-PAPP-A mice and in breast cancers overexpressing PAPP-A, the elevation in collagen during gestation amplifies the proteolytic activity of PAPP-A allowing excessive proliferative signaling pathways to be engaged (Fig 8B). However, the expression of STC1 and STC2 late during gestation and lactation inhibits PAPP-A activity.

When lactation is extended (Fig 8B, left panel), our results suggest that STC1 and STC2 reach high levels allowing inactivation of PAPP-A, even when it is overexpressed. This inhibition of PAPP-A during involution prevents the cleavage of IGFBP-5 and involution proceeds normally. In the absence of lactation or when lactation period is short (Fig 8B, right panel), PAPP-A levels exceed those of STC1 and STC2. As STC levels drop rapidly upon entry into involution, the excess PAPP-A promotes the degradation of IGFBP-5. Further, the abnormal expression of PAPP-A during involution allows the establishment of a positive feedback loop between collagen deposition and IGF signaling. Such constitutive proliferative signaling results in the formation of PABC shortly after pregnancy, and these tumors are characterized by TACS-3 signature and low IGFBP-5 (Fig 8B, right panel).

Adding to these effects is our finding that PAPP-A not only promotes further deposition of collagen but also favors the reorientation of collagen fibers into the pro-invasive TACS-3 signature. TACS-3 is an independent marker of worse prognosis and therefore may contribute to the aggressive nature of PABC. How PAPP-A favors the formation of TACS-3 is currently unclear. It is possible that since IGFBP-5 binds to collagen, the proteolysis of IGFBP-5 itself may alter the conformation of collagen. Alternatively, as collagen interacts with several proteoglycans and since PAPP-A contains a proteoglycan binding site, it is tempting to speculate that the proteolysis of proteoglycans may also contribute to the realignment of collagen fibers. In this setting, it is interesting that addition of collagen to Matrigel completely abolished the growth of control cells (Fig 3F). Collagen has been shown to have both growth-promoting (during involution) (Lyons *et al*, 2011) and growth-inhibiting (post-involution) (Maller *et al*, 2013) properties. Hence, since the experiment performed in Fig 3F mimics the post-involution condition, the result confirms that collagen is indeed growth inhibitory in the absence of PAPP-A expression.

Epidemiological studies of breastfeeding are complicated by the variability in the number of pregnancies, the period of time between pregnancies, variability of the length of breastfeeding following individual pregnancies as well as the concurrent use of bottle and breastfeeding. Considering this complexity, it is not surprising that several studies report contradictory findings. However, the result of a meta-analysis of the effect of breastfeeding on breast cancer risk revealed a clear protective effect of extended lactation

(Collaborative Group on Hormonal Factors in Breast C, 2002). The authors of this study suggested that the cumulative risk of breast cancer could be reduced by half should the period of breastfeeding be increased (Collaborative Group on Hormonal Factors in Breast C, 2002). Further, they found that breastfeeding is protective against more aggressive tumors (Faupel-Badger *et al*, 2013). Our results are in agreement with these observations and offer a potential mechanism that contributes to the protective effect of lactation. In addition, our data suggest that while extended lactation is protective against PAPP-A-mediated PABC, the oncogenic effect of PAPP-A during gestation is cumulative over multiple pregnancies and a single short lactation in the last pregnancy is sufficient to negate the protective effect of previous long lactation periods and promote the formation of PABC. Considering that women in modern society tend to reduce the length of breastfeeding, if our hypothesis is correct, these conditions would favor the development of PAPP-A-driven PABC. In light of these findings, it would be of interest to consider in the future whether the length of breastfeeding at each pregnancy affects the prevention of breast cancer as well as whether a correlation with PAPP-A-expression exists.

In our cohort of premenopausal patients, the overexpression of PAPP-A was observed in 79 % of cases. This is in contrast with the report that PAPP-A is overexpressed in all breast cancers (Mansfield *et al*, 2014). Reasons for this discrepancy include the method of detection by immunohistochemistry, scoring system as well as patient population analyzed since our study focused exclusively on premenopausal women while the other study focused exclusively in postmenopausal women. Alternatively, this result may indicate that an aging breast is more prone to express PAPP-A. Considering that tissue stiffness reduces with age, it is tempting to speculate that altered collagen structure in an aged breast may contribute to the apparent increased expression of PAPP-A in older women. This possibility will be tested in the future.

Collectively, data presented here identify PAPP-A as a pregnancy-dependent oncogene that promotes the formation of PABC and possibly identifies at the molecular level PABC as a distinct sub-type. Adding to this finding is the study by the Li's group indicating that oncogenes leading to maintenance of STAT5 even during involution contributes to the etiology of PABC (Haricharan *et al*, 2013). The identification of molecular drivers of PABC is essential in establishing a new classification, away from the current purely empirical definition of PABC as occurring within 1–2 years after pregnancy.

Lastly, since our results show that PAPP-A-positive PABC are more sensitive to anti-IGF therapy, and since several drugs targeting the IGF pathways are being tested in the clinic (Yang & Yee, 2012), our findings open the possibility of developing better targeted therapy for PABC in the future.

## Materials and Methods

### Animal procedures

Animal protocols were approved by the Institutional Animal Care and Use Committee (IACUC) of the Icahn School of Medicine at Mount Sinai. Mice overexpressing *PAPP-A* in the mammary gland (MMTV-PAPP-A) in a FVB/n background were generated as follows:

The PAPP-A wild-type cDNA was cloned into an expression plasmid containing the mouse mammary tumor virus (MMTV) long terminal repeat plus simian virus 40 intron and polyadenylation signals. A linearized MMTV-PAPP-A construct was microinjected into FVB/n mouse oocytes at the Mount Sinai Mouse Genetics Shared Research Facility, and both female and male founders were identified by genotyping tail genomic DNA. Animals were sacrificed and tissues were frozen at −80°C for biochemical analyses (for tissue analysis protocols, see "Whole-mount analysis" and "Histology and immunohistochemistry"). For the pregnancy time-course experiments, females ($n = 9$ mice; three mice per time point) in breeding cages were checked for vaginal plugs (day 0) and were sacrificed at days 3, 9, or 12 as described above. For the lactation time course, females ($n = 3$ mice) were sacrificed 12 days after dropping litter and feeding pups. For the involution time course, pups were separated from mothers 2 days or 2 weeks after birth and females ($n = 12$ mice; three mice per time point) were sacrificed at days 1, 3, 6, or 12. For hormone treatment, cervical region insertion of a 0.72 mg/ml 17 beta-estradiol pellet (Innovative Research of America, Cat#SE-121), daily peritoneal injection of 17 ng progesterone (Calbiochem, Cat#5241) in 100ul sterile PBS, or both estrogen and progesterone were administered in virgin MMTV-PAPP-A females ($n = 3$ mice per group). The animals were maintained for 1 week before harvesting mammary glands.

## RNA extraction and real-time RT–PCR

Total RNA was extracted from cell lines and mammary tissues using RNeasy Plus Mini kit (Cat#74134). Hundred nanograms of each triplicate sample was used in real-time RT–PCR using Takara One-Step PrimeScript RT–PCR kit (Cat#RR064A) following the manufacturer's protocols. Mouse PAPP-A primers (forward: 5′-TCCGCTCTTTCG ACAACTTT-3′ and reverse: 5′-CATGGTAGTGGTGGTTGCTGG-3′), human PAPP-A primers (forward: 5′-GTCAATGTTCCTTCCAGT GC-3′ and reverse: 5′-CTTGTGCTTATTCTCTCGGGC-3′), mouse STC1 primers (forward: 5′-CCCAATCACTTCTCCAACAGA-3′ and reverse: 5′-GAAGAGGCTGGCCATGTTG-3′), and mouse STC2 primers (forward: 5′-AGGAGAACGTCGGTGTGATT-3′ and reverse: 5′-CTGTTCACACTGAGCCTGGA-3′) were used along with the human beta-actin or mouse GAPDH primers as loading control.

## Whole-mount analysis

Mouse mammary glands were dissected, spread onto glass slides, and fixed in Carnoy's fixative (six parts absolute ethanol, three parts chloroform, one part glacial acetic acid) overnight. Mammary glands were placed in 70% ethanol for 15 min, with gradual exchange in MilliQ H$_2$O before incubation in carmine aluminum stain overnight. Slides were stored in 70% ethanol and imaged at 1× magnification under a light microscope.

## Histology and immunohistochemistry

Histological slides were prepared as 4-μm formalin-fixed sections embedded in paraffin and stained with H&E by the Oncological Sciences department histology core facility at Mount Sinai School of Medicine. For immunostaining of paraffin-embedded sections with IGFBP-5 (Santa Cruz, Cat#sc-13093) and pS473-Akt (Cell Signaling,

Cat#4060S), samples were deparaffinized in xylene and rehydrated in water. Antigen retrieval was performed in citrate buffer. Endogenous peroxidase activity was blocked with 3% hydrogen peroxide for 30 min. All buffer washes between incubations were with 50 mM Tris–HCl, pH 7.6. Primary antibody incubation was performed for 1 h at 1:50, RT and detected with the LSAB+ kit peroxidase (DAKO, Cat#K0690) with biotinylated anti-rabbit IgG for 1 h and streptavidin peroxidase for 10 min. Slides were developed with AEC (3-amino-9-ethyl-carbazole) chromogen (DAKO, Cat#K0690) for 10 min. Sections were then counterstained in hematoxylin and mounted with Permount (Fischer Scientific, Cat#SP15-100). For staining of sections with the PAPP-A D8-mIgG2a mouse monoclonal antibody (Mansfield *et al*, 2014, 1:260), samples were deparaffinized for 1 h at 65°C in an oven, with no antigen retrieval. Endogenous peroxidase activity was blocked with 3% hydrogen peroxide for 10 min. All buffer washes between antibody incubations were with 50 mM Tris–HCl + 0.1% Triton X-100, pH 7.6. Primary antibody incubation was performed for 1 h at 1:260, RT and detected with the LSAB+ kit peroxidase with biotinylated anti-rabbit IgG for 30 min and streptavidin peroxidase for 10 min. Slides were developed with AEC (3-amino-9-ethyl-carbazole) chromogen for 10 min. Sections were then counterstained in hematoxylin and mounted with Permount.

## Immunoblot

Mammary glands were homogenized in NP-40 lysis buffer (50 mM Tris, 250 mM NaCl, 5 mM EDTA, 0.5% NP-40, 50 mM NaF, 1 mM DTT) and 30 μg total protein in Laemmli buffer per sample was loaded on SDS–glycine polyacrylamide gels ran at 80 V for 30 min and 200 V for 45 min. Proteins were transferred to nitrocellulose membranes for 1 h at 80 V. Membranes were blocked in 4% milk in TBS-T and incubated on a rotator overnight at 4°C in the following primary antibodies: PAPP-A H-175 (Santa Cruz, Cat#sc-50518, 1:200), IGFBP-5 (Upstate, Cat#06-110, 1:1,000), actin (Chemicon Int'l, Cat#1378996, 1:20,000), phospho-Akt Ser 473 (Cell Signaling, Cat#4051S, 1:1,000), Akt (Cell Signaling, Cat#9272S, 1:1,000), phospho-STAT-5a/b (Millipore, Cat#05-495, 1:1,000), STAT-5 (1:1,000), STC1 (R&D Systems, Cat#AF2958, 1:1,000), STC2 (Santa Cruz, Cat#sc-14350, 1:200), and GAPDH (Calbiochem, Cat#CB1001, 1:20,000). After three washes in TBS-T, membranes were incubated in secondary rabbit or mouse antibodies coupled to peroxidase (Jackson Immunoresearch, Cat#115-035-146/111-035-146, 1:5,000) in 2% milk/TBS-T for 1 h, RT. Signal detection was performed with ECL (GE healthcare, Cat#RPN2106) following manufacturer's protocols.

## Masson's trichrome stain for collagen and analyses

Masson's trichrome staining (DAKO, Cat#AR173) of sections was performed following manufacturer's instructions. Triplicate images of stained sections were collected for representative areas of 2,560 × 1,920 pixels (0.44 μm/pxls). The blue stain was selected for and converted to black with Adobe Photoshop CS5's (version 12.0 × 64) magic wand tool. The images were opened in ImageJ (version 1.46r), and the black selections were quantified as % area positive for collagen. The images were normalized against the total epithelia in the image, in which the final ratio represents collagen/epithelia.

**Picrosirius red stain and circularly polarized light microscopy**

Paraffin-embedded sections were de-waxed and hydrated and then stained for 8 min with Weigert's hematoxylin. Sections were then stained for 1 h in Picrosirius red solution (1% Direct Red 80, Sigma Cat#365548, 1.3% Picric acid in water, Sigma Cat#P6744). Sections were washed in two changes of acidified water, dehydrated in three changes of 100% ethanol, and mounted with Permount. Triplicate images of each duplicate time point sections of mammary ducts were examined and taken with a light microscope equipped for use with circularly polarized light. The channels were split into red (thick collagen fibers) and green (thin collagen fibers) using Adobe Photoshop CS5 and quantified as % of total epithelial area.

**Second harmonic generation imaging of mouse samples and analyses**

All mouse mammary gland SHG images were captured on an Olympus FV1000MPE Fluoview multi-photon microscope (Tokyo, Japan) in the Microscopy CORE at the Icahn School of Medicine at Mount Sinai, supported with funding from NIH Shared Instrumentation Grant (1S10RR026639-01). Specs for imaging include a Coherent Chameleon Vision II Ti:S laser (Santa Clara, CA) with a 680- to 1,080-nm tuning range, dispersion compensation, and 140 fs pulse width. An Olympus XLPlanN 25×/1.05 numerical aperture water immersion lens was used for image acquisition. The excitation wavelength was set at 900 nm, and signal collection was performed by backwards imaging with a 420- to 460-nm band-pass filter, a 485 dichroic mirror (GR/XR filter cube, Olympus), and an external detector. Images were acquired at a 1,024 × 1,024 pixel resolution while keeping the power constant throughout. Collagen intensity versus distance analysis was performed in six replicate ducts per mouse in three mice per group (NT versus PAPP-A, $n = 18$ ducts per group) following a protocol described previously (Lyons *et al*, 2011). Involution duct SHG images were opened in ImageJ (version 1.46r), and a rectangular selection of 50 × 100 pixels was made bordering the edge of ducts in four directions. The four regions of interest (ROI) were analyzed using the profile plot function, and intensity of the SHG signal over a 40-μm distance for the four ROIs was averaged as the signal for a single duct. For curvelet orientation analyses of SHG images in mice, a 3 × 3 stitched image of 1,024 × 1,024 pixel per tile (final image of 3,072 × 3,072) were captured in triplicates for five representative MMTV-PAPP-A tumors. The collagen fiber (curvelets) angle against the selected tumor border (ROI, average of 293.44 curvelets) from each of the triplicate images per tumor was analyzed using the CurveAlign (version 2.2 R2012a) software. The curvelet angle distribution output ($< 40°$, $40° - 60°$, or $> 60°$) from the software was utilized.

**Establishment of MCF7^PAPP-A stable clones**

MCF7 breast cancer cells were grown in Dulbecco's modified Eagle's medium (DMEM) supplemented with 10% fetal bovine serum. MCF-7^PAPP-A stable clones were generated by transfection of a PAPP-A wild-type (pcDNA3.1-PAPP-A) construct using Mirus (TransIT-LT-1, Mirus) following manufacturer's instructions.

G418-selected stable clones were confirmed for PAPP-A expression by qRT–PCR (see "RNA Extraction and Real-Time RT–PCR") and Quantikine PAPP-A ELISA (R&D systems, Cat#DPPA00) following manufacturer's protocol.

**PAPP-A protease assays**

Collagen gels were formed in Eppendorf tubes at 37°C for 30 min by mixing 1:1 rat tail collagen I protein (Gibco, Cat#A1048301) with neutralizing solution (100 mM Hepes, PH7.3 in 2× PBS). Similarly, laminin (Life Technologies, Cat#23017-015) gels at a final concentration of 50 ng/μl were formed at 37°C for 30 min. Cell-free protease assays were performed using PAPP-A protein secreted in 24-h serum-free culture media from MCF-7^PAPP-A stable clones grown to confluency (in complete media) on 10-cm plates. Culture media from parental MCF-7 were used as a negative control for PAPP-A protein in cell media. Fifty nanograms of rIGFBP-5 was co-incubated with 15 μl laminin or collagen gel at 37°C for 3 h in either 15 μl MCF7 parental media or MCF-7^PAPP-A media. The reaction was inactivated by addition of 1× Laemmli sample buffer to 40 μl final volume and boiling for 5 min. *In vitro* cleavage of endogenous IGFBP-5 from cell lines was performed by growing MCF-7 parental or MCF-7^PAPP-A on plastic or collagen gels for 3 days. Cells were harvested in 2× lysis/RIPA buffer (50 mM HEPES, PH7.4, 150 mM NaCl, 2 mM EDTA, 2 mM NaF, 2% NP-40, protease inhibitor cocktail) and prepared for immunoblot in 1× Laemmli sample buffer to 40 μl final volume and boiling for 5 min. PAPP-A activity from mouse mammary gland was assessed as follows: mammary gland extracts were prepared in non-reducing NP40 lysis buffer. Hundred micrograms of the extract was co-incubated with 17 ng rIGFBP-5 at 37°C for 18 h. The reaction was inactivated by addition of 5× Laemmli sample buffer and boiling for 5 min.

**Mouse xenografts and drug treatment**

The week before tumor cell injection, a 0.72 mg/ml 17 beta-estradiol pellet (Innovative Research of America, Cat#SE-121) was inserted in the cervical region of female nude mice. 2.5 million cells (MCF-7 parental or MCF-7^PAPP-A) in 100 μl of a Matrigel/collagen 1:1 mixture (BD Biosciences, Cat#354234; Gibco, Cat#-A1048301) were injected in the right and left inguinal mammary fat pads of virgin mice ($n = 4$ mice, eight tumors total). Palpable tumors were measured with a caliper starting 6 days after injection and three times a week for 3 weeks. Tumors were harvested for histological analyses or as RNA and protein samples at −80°C. In the following experiments with pregnancy, every female ($n = 3$ mice, six tumors total) was monitored and injections/measurements were made according to when each female dropped litter: In the "virgin versus involution" xenografts, each female in the involution group was injected in the right and left inguinal mammary fat pads with tumor cells resuspended in 100 μl of Matrigel upon separating pups (start of involution). In the "involution only" xenografts, pups were separated at birth and tumor cells were injected immediately after. In the "lactation only" group, tumors were injected immediately after litter was dropped. Pups were kept with mothers for the duration of the xenograft. In the "lactation + involution" group, pups were removed after 2 weeks

of lactation, and tumor cells were injected immediately after removal of litter. Palpable tumors were measured with a caliper two times a week for 2 weeks. Tumors were collected as mentioned above. For injection of IGF-1R inhibitor, PQ401 (ENZO Life Sciences, ALX-270-459-M005) was made to a stock concentration of 25 mg/ml in 50% polysorbate 80/50% ethanol. The stock was diluted with sterile phosphate-buffered saline to make an injection stock at 4 mg/ml (freshly made before each injection). Mice with palpable tumors (15 days after tumor injection) in the involution group ($n = 3$ mice, five tumors total) received injections of 250 μl injection stock every 2 days, three times total. Tumor volume was measured with calipers and recorded at the indicated time points.

### Immunodepletion

Ten micrograms of control goat IgG (Santa Cruz), STC1 (R&D Systems), or STC2 (Santa Cruz) antibodies was incubated for 1 h at 4°C with 25ul prewashed protein A agarose beads (Santa Cruz) in 200 μl DTT-free NP-40 (reaction buffer) with 200 μg/ml BSA. Beads were washed three times in reaction buffer and mixed with 300 μg of protein extract in reaction buffer (100 μl). The reaction was incubated for 2 h at 4°C. For immunodepletion, the samples were spun down and supernatant was collected for subsequent experiments. One-third of the supernatant fraction was taken for the PAPP-A protease assay (100 μg extract/assay), one-third for assessing STC1 or STC2 protein levels, and one-third for assessing PAPP-A protein levels by ELISA following manufacturer's protocol (R&D systems, Cat#DPPA00).

### Collection of patient samples

Collection of pregnancy history and tissue sections from newly diagnosed breast cancer patients was performed according to our IRB-approved protocol. Chat data as recorded by clinicians include menopause status, age and number of pregnancy, years since last pregnancy, and hormone receptor status. Histological sections from patients were obtained from the Department of Pathology at Mount Sinai.

### Scoring of human breast cancer specimen

Multiphoton microscopy for patient specimens was performed at the University of Wisconsin Laboratory for Optical and Computational Instrumentation (Conklin *et al*, 2011). Imaging was done on a Prairie Technologies Ultima IV multiphoton microscope at an excitation wavelength of 890 with a Spectra Physics Insight laser. All images were collected with 20× 1.0 NA Objective (Olympus) and using a 445-nm bandpass filter to discriminate for SHG collagen. SHG images at a resolution of 1,024 × 1,024 pixels were taken in triplicates, and two tumor border ROIs per image were analyzed using the CurveAlign (version 2.2 R2012a) software, covering 700–1,000 collagen fibers (curvelets) analyzed over 6 ROIs per sample. Curvelets with an angle against the border below 40° were assigned a score of 1, between 40° and 60° a score of 2, and above 60° a score of 3. The fraction of curvelets belonging to each of these three angle ranges was multiplied by the respective scores and added to obtain the final TACS-3 score per patient. $n = 18$ for

nulliparous, $n = 28$ for parous. For PAPP-A and IGFBP-5 immunohistochemistry, a score of 0 was assigned to an internal negative control stain in invasive regions, score 1 to regions with weak staining, and score 2 to regions with strong staining. A section was considered positive if more than 20% of cells displayed a score 2 stain.

### Statistical analysis

All statistical analyses were performed using GraphPad Prism version 6. Specific statistical tests are listed in the legend of figures for each experiment. No statistical method was used to predetermine sample size. The investigators were not blinded to allocation during experiments and outcome assessment.

**Expanded View** for this article is available online.

## Author contributions

YT is the PhD student who as conducted the vast majority of the experiments. CO has provided PAPP-A plasmids and antibody and established the immunohistochemistry protocol for PAPP-A. CN is a breast pathologist who performed the scoring of the immunohistochemistry. KA is a breast oncologist who has consented patients and obtained their history of pregnancy. SJ is a pathologist who had collected blocks of patients diagnosed with breast cancer shortly after pregnancy. HS is a breast surgeon who has identified premenopausal patients who never had children and conducted the chart review. PJK defined the TACS-3 signature and helped with the interpretation of the data. KWE has provided training in SHG. JM is a biostatistician who has provided his expertise in the statistical analysis performed in this study. DG is the PI of the laboratory, designed the experiments, interpreted the data, and wrote the manuscript.

## Conflict of interest

The authors declare that they have no conflict of interest.

---

### The paper explained

**Problem**

Pregnancy is associated with increased risk of breast cancer, but the link between pregnancy and breast cancer remains unclear. Further, breastfeeding is protective against breast cancer, but how this protective effect is mediated is also unknown.

**Results**

Results described in this manuscript propose that the abnormal presence of the protease PAPP-A in a breast during pregnancy promotes breast cancer. However, we found that inhibitors of PAPP-A are present during lactation and that long lactation protects mice from developing cancer even when they show the abnormal presence of PAPP-A in their mammary glands. Data obtained in mice was validated in patients based on their history of pregnancy.

**Impact**

The clinical impact of this work is the identification of the first marker of pregnancy-associated breast cancer and a mechanism to explain why breastfeeding is protective against breast cancer. Further, since drugs targeting PAPP-A-induced pathways already exist, the results may lead to targeted treatment for young mothers suffering from pregnancy-associated breast cancer.

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
