## [Review Process File · EMBO Molecular Medicine]

Lactation opposes pappalysin-1-driven pregnancy-associated breast cancer

Yukie Takabatake, Claus Oxvig, Chandandeep Nagi, Kerin Adelson, Patricia Keely, Kevin Eliceiri and Doris Germain

Corresponding author: Doris Germain, Icahn School of Medicine at Mount Sinai

Review timeline:

Original Submission Date:	30 July 2015
Editorial Decision:	02 September 2015
Revision received:	02 December 2015
Editorial Decision:	14 January 2016
Resubmission received:	03 February 2016
Accepted:	10 February 2016

Editor: Roberto Buccione

Transaction Report:

1st Editorial Decision

02 September 2015

Thank you for the submission of your manuscript to EMBO Molecular Medicine. We have now heard back from the three Reviewers whom we asked to evaluate your manuscript.

Although the Reviewers agree on the potential interest of the manuscript, the issues raised are of a fundamental nature. I will not dwell into much detail, but I would like to highlight the main points.

Reviewer 2 expresses two main and important concerns. The first is that s/he is not satisfied with the quality and depth of analysis of the human sample dataset, including a clear disagreement on the definition of the premenopausal patients; this is not merely a formal issue as there are direct implications on the main conclusions. The other point is that the number of mice used appears insufficient to claim statistical significance. This Reviewer also lists other important items of concern that require your action.

Reviewer 3 notes that the physiological role of PAPP-A is far from proven based on the experimentation. S/he is also concerned that a possible role of progesterone in driving transgene expression has not been excluded. These concerns are of great importance for us as they impinge on the most interesting potential messages of the manuscript. Reviewer 3 also laments issues with statistics and numbers of experimental animals, and lists a number of other relevant points.

Reviewer 1 is less reserved but also raises the issue of the physiological role of PAPP-A. S/he also provides an extensive list of items that need your action, which all appear important but feasible and should be addressed in full.

In conclusion, while publication of the paper cannot be considered at this stage, given the potential interest of your findings and after internal discussion, we have decided to give you the opportunity to address the criticisms.

We are thus prepared to consider a substantially revised submission, with the understanding that the Reviewers' concerns must be addressed with additional experimental data where appropriate and that acceptance of the manuscript will entail a second round of review. The overall aim is to significantly upgrade the relevance and conclusiveness of the dataset, which of course is of paramount importance for our title.

***** Reviewer's comments *****

Referee #1 (Comments on Novelty/Model System):

The paper addresses a highly relevant topic in breast cancer research, pregnancy-associated breast cancer (PABC), and presents a mechanism linking several parameters associated or already implicated in pregnancy-associated breast cancer. In this regard, it is easy to misread the main message of the manuscript, which actually goes further by aiming to provide a mechanism how extended lactation protects against PABC, meriting the high medical impact and novelty. There are a few small technical and formal issues with an otherwise highly interesting manuscript, formalised below,

Referee #1 (Remarks):

The title of the manuscript, "Lactation opposes Pappalysin-1 driven pregnancy-associated breast cancer" aptly describes its main message. By transgenic expression of the secreted protease Pappalysin-1 (PAPP-A) under control of the mouse mammary tumor virus (MMTV), the authors build a mechanism whereby increased expression of PAPP-A, together with collagen, causes a delay in involution through enhanced cleavage of IGFBP-4 and -5. It is implied that this, in turn, leads to increased IGF-signaling, although this is not conclusively shown. The authors link this delay in involution with the appearance of tumors in a significant fraction of parous, but not virgin glands of PAPP-A transgenic animals. Strikingly, this pregnancy-associated tumor formation is completely controlled by the duration the mice are allowed to breast-feed their offspring: breast-feeding for a period of 2 weeks protects from tumor formation. The authors argue that this is due to increased expression of inhibitors of PAPP-A, STC1 and STC2.

Overall, the manuscript is very well written and highly original. The experiments are carefully executed and the figures comprehensive and beautifully crafted. The methods section provides excellent detail. The topic is of high clinical relevance, as the authors delineate clearly in the discussion: first of all, the data suggest that pregnancy-associated breast cancer (PABC) might encompass a distinct subgroup of breast cancer. More importantly, the well-recognized increased risk for postpartum development of breast cancer might be mitigated by extended periods of lactation. Although the clinical data the authors present fall somewhat short of supporting this claim, the experimental data are highly convincing and clearly delineate a possible molecular mechanism.

Specific Points to address:

Figure 1

How do the PAPP-A secreted levels differ during involution between NT control and PAPP-A transgenics? It is striking that the authors do not provide more comparison of the fold-differences during the different phases they analyse, i.e. virgin, pregnancy, lactation and involution. Figure 1C could contain IHC for PAPP-A and Figure 1D a western blot. One would expect to observe an anti-proportional correlation between PAPP-A levels and IGFBP-5 levels, strengthening the mechanism the authors propose.

Figure 2

The Figure assembly is at first glance a bit confusing. It would be useful to include graph and image titles that clearly delineate which images are quantified.

Figure 2D - the contrast setting of the NT control at day 1 looks completely different than all the other images. The authors should make sure the same settings were used to take that image.

Figure 3

Figure 3F: it is somewhat surprising that the control tumors do not grow at all, even though the

authors injected them with a mix of collagen:matrigel - this should be discussed.

Figure 3G: where is the control lysate coming from if there is hardly any tumor to be harvested? Is there still lysate available? In that case, it would be nice to include p-AKT to link the decreased IGFBP-5 levels to IGF-signaling.

Figure 3J: While this is an acceptable presentation, why are there no growth curves depicted, as in Figure 3H and 3I? How exactly was the relative growth rate calculated?

Figure 4

Figure 4A, B: To strengthen the claim that multiple pregnancies compound the pro-tumorigenic effect of PAPP-A overexpression, it would be useful to provide quantification of the difference in tumor burden in mice examined after 1 or 3 pregnancies. Maybe this would be possible by counting the apparent lesions per field using the whole-mounts.

Figure 4D: It would be useful to aim for a bit more characterization of the PAPP-A tumors. Maybe a breast pathologist could comment on invasiveness and angiogenesis as parameters of aggressiveness. This could also be done in comparison to the HER2 and Wnt tumors to further strengthen the claim, that PAPP-A tumors are distinct. It would also be quite interesting to perform IHC for p-AKT to determine whether Akt-signaling is activated in PAPP-A tumors, possibly as a consequence of elevated IGF-1 signaling.

Figure 5

Figure 5A, B: As discussed above, a blot and IHC for p-AKT would greatly strengthen the authors proposed mechanism. It might also help to better define the role of newly deposited, fibrillar collagen in this process by tightly correlating PAPP-A activity with pro-oncogenic signaling.

Figure 5E,G: Why are there no blots for total AKT and Stat5a/b? Also, for AKT, it should be indicated which phosphorylation sites are examined.

Figure 6

Figure 6A: It would be highly useful, for the entire section, to clarify what exactly the time differences between short and full lactation are. This is somewhat difficult to glean from the text and is also missing in the figure.

It might be a good idea to density-quantify the western blots, particularly 6J and 6K.

Figure 8

Figure 8 depicts the proposed mechanism on how lactation influences PAPP-A driven tumorigenesis beautifully. However, it also highlights the only weakness of the manuscript: elevated Papp-A levels lead to an extended period of involution with increased IGF-1 signaling. However, the latter part is merely deduced from a decrease in IGFBP-4 and -5 levels. That is ok, no manuscript can show everything. But as discussed above, maybe it would be possible, through a few added analyses, to strengthen the latter point.

Author contributions:

Amazingly, the first author forgot to include herself. Too much humility!

Referee #2 (Remarks):

Comments

The manuscript titled "Lactation opposes Pappalysin-1 driven pregnancy-associated breast cancer" investigates the role of pappalysin-1 (PAPP-A) as a pregnancy-dependent oncogene. It proposes very clearly the linkage between the protease PAPP-A, its substrates insulin-like growth factor binding protein (IGFBP)-4 and -5 and altered collagen conformations as a molecular mechanism leading to pregnancy-associated breast cancer (PABC) dependent on lactation. To proof this cohesion the authors utilized a PAPP-A transgenic mouse model and transferred their findings to human PABC by analyzing tissue sections from newly breast cancer diagnosed pre-menopausal patients.

The mode of presentation is very clear and comprehensible. The subdivisions are chosen reasonably resulting in a structured, didactic valuable manuscript.

Major concern:

Regarding analysis of human samples:

The classification of pre-menopausal patients ("diagnosed with breast cancer before or at the age of 51") is inaccurate. A pre-menopausal state is defined by regular monthly cycles, a plasma 17 β -estradiol level of 35 to 400 pg/ml or a plasma FSH-level of 25.8 - 134.8 mIU/ml. A post-menopausal state is defined by more than a year of absence of the menstrual period, a plasma 17 β -estradiol level of < 35 pg/ml or a plasma FSH-level of 25.8 - 134.8 mIU/ml. As the patient population consists of newly diagnosed breast cancer cases, these information should be available at least for therapeutic decisions and should be included for the classification of the pre-menopausal state. This is crucial as the authors emphasize several times to focus on an exclusively pre-menopausal collective, which cannot be assumed on the basis of their classification.

- Fig. 2A: 2 mice per time point is a very low number to proof statistical significance. In cases of very low numbers it would more appropriate to choose the median instead of the mean value as it is not to expect that the small population have a normal distribution.

Minor concerns:

Regarding authors guidelines:

- The total character count is not indicated on the title page and exceeds the demanded maximum of 60.000 characters.
- The postal codes of the cities of the corresponding institutes are missing.
- The institutional and/or licensing committee approving the animal experiments is not indicated.
- Keywords are missing.
- References are not indicated in the text like demanded by the authors guidelines. The References indicated in the bibliography do not suit the demanded citation style.

Regarding the mouse model:

- Fig. 1E: Typing error: It has to be change not "cahnge".
- Fig. 2C/E: Do these 6 representative areas derive from one mammary gland of one animal or from different glands/animals?
- Fig. 2G: Statistical analysis is missing or it has to be indicated, that here was no statistical analysis done.
- Fig. 3C/D: Housekeeping protein is not shown.
- Fig. 3J: Why is $p=0,01$ declared as not statistically significant?
- Fig. 4C and the H&E-section of Fig. 4D do not support the results given in the text. They represent expendable information and maybe should be shifted to the supplementary figures.
- Fig. 4G/H: Do the ROI derive from on mammary gland of one animal or from different glands/animals?
- Fig. 5A: Western Blot image of IGFBP-4 is cloudy compared to IGFBP-5.
- Fig. 6O/P/Q/R: The numbering has to be adapted, to ensure a congruent order in the running text of the results.

Regarding analysis of human samples:

The quality of the human data stands in contrast to the excellent quality of the transgenic mouse model and lacks to transfer these convincing results to human PABC. Due to the well established methodology it is conceivable to adapt its quality by expanding the number of included patients, by collecting the required additional information on pregnancy and lactation anamnesis and by correcting the current classification of pre-menopausal state as described above.

- Unfortunately there is no data available on lactation anamnesis and surprisingly there is no information on the number of pregnancies within the group of parous women.
- The number of patients included is low.

Regarding statistical analysis

- There was no normality test applied before statistical analysis.
- In cases of small numbers the utilization of median and consequently the utilization of the Mann-Whitney-test for comparison of two groups is recommended.

Referee #3 (Remarks):

EMM-2015-05692

Title: Lactation opposes pappalysin-1 drive pregnancy-associated breast cancer.

Corresponding author: Doris Germain

This manuscript addresses two clinically important but understudied areas of pregnancy-related breast cancer, namely the mechanisms for increased BC risk following pregnancy and the protective effect of lactation. The breadth of work is ambitious, and the questions fundamental with novel observations. Specifically, using a transgenic model with PAPP-A targeted to the mammary epithelium, the authors identify a role for the protease pappalysin-1 (PAPP-A) in collagen deposition, which in turn, facilitates the proteolysis of IGFBP-4 and -5. In a xenograft model comparing control to PAPP-A overexpressing MCF-7 cells, PAPP-A overexpressing cells form larger tumors with decreased IGFBP-5 levels, implicating the release of IGF as a primary mechanism for the observed increased tumor size. Further, the PAPP-A driven tumors have elevated tumor associated collagen (called TAC3), providing a plausible mechanism for the high metastatic rate of PABC observed in women. Importantly, lactation greater than 2 days in length inhibits PAPP-A driven tumor progression by upregulating expression of the putative PAPP-A inhibitors STC1 and STC2. In the PAPP-A transgenic mice, multiple rounds of pregnancy drive hyperplasia, suggesting PAPP-A as a pregnancy associated oncogene.

The links between PAPP-A, collagen and IGFBP cleavage are novel and provide important contributions to the breast cancer/breast density/collagen field. The lactation data are also highly novel. Further, the inclusion of collagen TAC assessment in a small cohort of parous and nulliparous breast cancer cases is supportive of the paper's primary hypothesis-and is a commendable addition to the manuscript.

Major Concerns

1. The authors rely on overexpression models and do not show a physiologic role for PAPP-A in normal pregnancy, lactation or weaning-induced involution. If the PAPP-A/collagen pathway is causally implicated in the initiation of postpartum breast cancers, upregulation with pregnancy is anticipated.
2. A key point in the paper is that PAPP-A is a pregnancy associated oncogene, as multiple pregnancies result in mammary hyperplasia. The authors clearly state that the MMTV-construct utilized to express PAPP-A is not hormone responsive, thus eliminating the trivial explanation that hyperplasia develops due to hormone regulated transgene expression. However, the authors have not ruled out the role of progesterone (or combined estrogen and progesterone at pregnancy levels) in driving transgene expression. Without these data, it is not possible to conclude that the increase in hyperplasia with pregnancy is not an artifact of the model (albeit, one that yields informative data on collagen regulation). This concern becomes more relevant given that the authors do not demonstrate a role for PAPP-A in normal pregnancy.
3. Given points 1 & 2, in the manuscript's current form, the data best support a role for PAPP-A in mediating collagen deposition and IGF bioavailability, independent of pregnancy, lactation or involution.
4. The roles of STC1 and STC2 in mediating the protective effect of lactation are not well developed. Differences in expression/function are expected between the short and full lactation models, but these data are not provided. Currently the data are correlative, and causal roles for STC1 and SCT2 are lacking. The wording of the conclusions should be tempered to reflect the correlative association.

Figures:

Fig 3F compared to Fig 3H, can the authors speculate as to why the addition of collagen to vector-only tumor cells would induce tumor loss? Is this observation robust/replicable?

Fig 5 F, is hyperplasia observed after the 3rd pregnancy in WT mice?

In many of the figures, the axis font is too small to read.

Methods Questions:

For most of the experiments, it is unclear how many mice were used per group as well as how many

replicate studies were performed. Further, error bars are missing from many of the graphs, suggesting absence of true replica data.

For Fig 7, please provide clinical parameters of tumor stage and size, as these parameters could independently influence TAC formation. Also, how many different areas within each tumor were assessed for TACS? What is the intra-tumor variation in TACS?

General:

The cited literature could be updated to include 1) work by others investigating pregnancy-dependent oncogenes using oncogene delivery that is demonstrably hormone independent (Yi Li lab) and literature/discussions on the time frame after pregnancy for which a diagnosis of pregnancy-associated breast cancer is considered, as substantial clinical data supports an extended definition (Albrektsen G., Lambe, M., and Borges, V).

1st Revision - authors' response

02 December 2015

Referee #1:

The paper addresses a highly relevant topic in breast cancer research, pregnancy-associated breast cancer (PABC), and presents a mechanism linking several parameters associated or already implicated in pregnancy-associated breast cancer. In this regard, it is easy to misread the main message of the manuscript, which actually goes further by aiming to provide a mechanism how extended lactation protects against PABC, meriting the high medical impact and novelty. There are a few small technical and formal issues with an otherwise highly interesting manuscript, formalised below,

Referee #1 (Remarks):

The title of the manuscript, "Lactation opposes Pappalysin-1 driven pregnancy-associated breast cancer" aptly describes its main message. By transgenic expression of the secreted protease Pappalysin-1 (PAPP-A) under control of the mouse mammary tumor virus (MMTV), the authors build a mechanism whereby increased expression of PAPP-A, together with collagen, causes a delay in involution through enhanced cleavage of IGFBP-4 and -5. It is implied that this, in turn, leads to increased IGF-signaling, although this is not conclusively shown. The authors link this delay in involution with the appearance of tumors in a significant fraction of parous, but not virgin glands of PAPP-A-1 transgenic animals. Strikingly, this pregnancy-associated tumor formation is completely controlled by the duration the mice are allowed to breast-feed their offspring: breast-feeding for a period of 2 weeks protects from tumor formation. The authors argue that this is due to increased expression of inhibitors of PAPP-A, STC1 and STC2.

Overall, the manuscript is very well written and highly original. The experiments are carefully executed and the figures comprehensive and beautifully crafted. The methods section provides excellent detail. The topic is of high clinical relevance, as the authors delineate clearly in the discussion: first of all, the data suggest that pregnancy-associated breast cancer (PABC) might encompass a distinct subgroup of breast cancer. More importantly, the well-recognized increased risk for postpartum development of breast cancer might be mitigated by extended periods of lactation. Although the clinical data the authors present fall somewhat short of supporting this claim, the experimental data are highly convincing and clearly delineate a possible molecular mechanism.

Specific Points to address:

Figure 1

"How do the PAPP-A secreted levels differ during involution between NT control and PAPP-A transgenics? It is striking that the authors do not provide more comparison of the fold-differences during the different phases they analyse, i.e. virgin, pregnancy, lactation and involution. Figure 1C could contain IHC for PAPP-A and Figure 1D a western blot. One would expect to observe an anti-

proportional correlation between PAPP-A levels and IGFBP-5 levels, strengthening the mechanism the authors propose.”

Response: The fold difference between the non-transgenic and transgenic is now shown in supplementary figure 1A and is discussed in the text. The fold difference in the expression of PAPP-A transgenic during different phases is also shown in supplementary figure 1B. As suggested, we have included IHC to match the data in figure 1C. This data is the new figure 1D. As for the western blot, since PAPP-A is a very high molecular weight protein and extracts from mammary gland make this analysis very difficult. However, in the light of the IHC of PAPP-A, we trust the data supports the expression of PAPP-A during this time course. This data shows an anti-proportional correlation between PAPP-A and IGFBP-5 during involution.

We also included a new section in the text describing the expression pattern of endogenous PAPP-A during the various phases and this data is now shown in supplementary figure 5C. This data shows that endogenous PAPP-A is not expressed during involution and therefore is also consistent with elevated levels of IGFBP-5 during involution.

Figure 2

“The Figure assembly is at first glance a bit confusing. It would be useful to include graph and image titles that clearly delineate which images are quantified“

Response: Graphs of quantification were added next to the images that are quantified. This is also clarified in the legend to the figure.

Figure 2D –

“the contrast setting of the NT control at day 1 looks completely different than all the other images. The authors should make sure the same settings were used to take that image“

Response: The contrast setting in the NT control at day 1 has been rectified.

Figure 3

“Figure 3F: it is somewhat surprising that the control tumors do not grow at all, even though the authors injected them with a mix of collagen:matrigel - this should be discussed“

Response: Collagen has been shown to have both growth-promoting (during involution) and growth-inhibiting (post-involution) properties (Lyons et al., Nat. Med., 2011, Maller et al., J. Cell Science, 2013). Hence, since the experiment performed in figure 3F mimics the post-involution condition, the result suggests that collagen is indeed growth-inhibitory for the control cells. However, when PAPP-A is expressed, collagen enhances its oncogenic function. Discussion of this effect was added to the discussion page 18 first paragraph.

“Figure 3G: where is the control lysate coming from if there is hardly any tumor to be harvested? Is there still lysate available? In that case, it would be nice to include p-AKT to link the decreased IGFBP-5 levels to IGF-signaling“

Response: The control tumors are indeed small but while they did not grow, they still contain 2×10^6 cells allowing their analysis by Western blot. Phospho-AKT has been added to figure 3G.

“Figure 3J: While this is an acceptable presentation, why are there no growth curves depicted, as in Figure 3H and 3I? How exactly was the relative growth rate calculated? “

Response: The graph in figure 3J has been changed to resemble those shown in 3H and I. Unlike 3H and I where there is no treatment, for 3J drug treatment begun at 17 days after injection of cancer cells. Therefore, the starting tumor volume is different between the groups and as expected the PAPP-A tumors are larger. We now show that the slope of the growth curve before treatment is 72% in the PAPP-A groups but of only 31% in the control group. After treatment the slope reduced to 22.8% in the PAPP-A groups but was 39% in the control group. Therefore, the effect of PQ401 is significant in the PAPP-A group but not in the control group.

Figure 4

“Figure 4A, B: To strengthen the claim that multiple pregnancies compound the pro-tumorigenic effect of PAPP-A overexpression, it would be useful to provide quantification of the difference in tumor burden in mice examined after 1 or 3 pregnancies. Maybe this would be possible by counting the apparent lesions per field using the whole-mounts”

Response: This quantification was added to figure 4B.

“Figure 4D: It would be useful to aim for a bit more characterization of the PAPP-A tumors. Maybe a breast pathologist could comment on invasiveness and angiogenesis as parameters of aggressiveness. This could also be done in comparison to the HER2 and Wnt tumors to further strengthen the claim, that PAPP-A tumors are distinct. It would also be quite interesting to perform IHC for p-AKT to determine whether Akt-signaling is activated in PAPP-A tumors, possibly as a consequence of elevated IGF-1 signaling”

Response: Dr. Shabnam Jaffer who is a breast pathologist and now an author, has performed this analysis. She found that while the PAPP-A tumors are adenocarcinoma with metaplastic features, the pathologies of the Her and Wnt tumors in mice do not share a human equivalent. The description of the pathology of the PAPP-A tumors was added to the text (page 9). We also performed p-AKT IHC on the PAPP-A tumors and lesions and this data was added to figure 4D.

Figure 5

“Figure 5A, B: As discussed above, a blot and IHC for p-AKT would greatly strengthen the authors proposed mechanism”

Response: A western blot of p-Akt as well as IHC was added to figure 5A.

“Figure 5E,G: Why are there no blots for total AKT and Stat5a/b? Also, for AKT, it should be indicated which phosphorylation sites are examined”

Response: Blots of AKT and phospho- AKT were added to figure 5E and G. In addition, in all blots of phospho-Akt, phosphor-serine 473 is now shown.

Figure 6

“Figure 6A: It would be highly useful, for the entire section, to clarify what exactly the time differences between short and full lactation are. This is somewhat difficult to glean from the text and is also missing in the figure.”

Response: The short lactations are less than 2 weeks ranged from 0 days to 13 days, with an average of 1.076 days. Long lactations are more than >2 weeks and ranged from 21 days to 24 days, with an average of 21.25 days. This clarification was added to the legend of figure 6A as well as in the text.

“It might be a good idea to density-quantify the western blots, particularly 6J and 6K”

Response: Quantifications were added to these westerns, which are now shown in 6I as a result of the reorganization of figure 6 at the request of critique 2.

Figure 8

“Figure 8 depicts the proposed mechanism on how lactation influences PAPP-A driven tumorigenesis beautifully. However, it also highlights the only weakness of the manuscript: elevated Papp-A levels lead to an extended period of involution with increased IGF-1 signaling. However, the latter part is merely deduced from a decrease in IGFBP-4 and -5 levels. That is ok, no manuscript can show everything. But as discussed above, maybe it would be possible, through a few added analyses, to strengthen the latter point.”

Response: We agree and the added analyses of western blots and IHC of phospho-AKT have strengthened the contribution of IGF-signaling. We appreciate the reviewer's valuable input and suggestions for experiments.

Author contributions:

Amazingly, the first author forgot to include herself. Too much humility!

Response: This was rectified. Thank you.

Referee #2 (Remarks):

Comments

The manuscript titled "Lactation opposes Pappalysin-1 driven pregnancy-associated breast cancer" investigates the role of pappalysin-1 (PAPP-A) as a pregnancy-dependent oncogene. It proposes very clearly the linkage between the protease PAPP-A, its substrates insulin-like growth factor binding protein (IGFBP)-4 and -5 and altered collagen conformations as a molecular mechanism leading to pregnancy-associated breast cancer (PABC) dependent on lactation. To proof this cohesion the authors utilized a PAPP-A transgenic mouse model and transferred their findings to human PABC by analyzing tissue sections from newly breast cancer diagnosed pre-menopausal patients.

The mode of presentation is very clear and comprehensible. The subdivisions are chosen reasonably resulting in a structured, didactic valuable manuscript.

Major concern:

"Regarding analysis of human samples:

The classification of pre-menopausal patients ("diagnosed with breast cancer before or at the age of 51") is inaccurate. A pre-menopausal state is defined by regular monthly cycles, a plasma 17 β -estradiol level of 35 to 400 pg/ml or a plasma FSH-level of 25.8 - 134.8 mIU/ml. A post-menopausal state is defined by more than a year of absence of the menstrual period, a plasma 17 β -estradiol level of < 35 pg/ml or a plasma FSH-level of 25.8 - 134.8 mIU/ml. As the patient population consists of newly diagnosed breast cancer cases, these information should be available at least for therapeutic decisions and should be included for the classification of the pre-menopausal state. This is crucial as the authors emphasize several times to focus on an exclusively pre-menopausal collective, which cannot be assumed on the basis of their classification "

Response: That information was indeed available in the patient charts and we do agree that our previous classification of pre-menopausal patients was not appropriate. Using the clinical definition of pre-menopausal status, a few additional pre-menopausal patients were identified (previously not included). The correct classification of pre-menopausal status (as obtained from charts review) also applies to the additional patients that were added to increase sample size.

- "Fig. 2A: 2 mice per time point is a very low number to proof statistical significance. In cases of very low numbers it would more appropriate to choose the median instead of the mean value as it is not to expect that the small population have a normal distribution"

Response: Figure 2A was moved to supplementary figure 2. We have added an additional mouse per time points and so therefore, all time points are now the results of the analysis of 3 mice. Also it should be noted that there is a total of 4 time points leading to 12 mice per group.

Minor concerns:

Regarding authors guidelines:

“- The total character count is not indicated on the title page and exceeds the demanded maximum of 60.000 characters.

- The postal codes of the cities of the corresponding institutes are missing.
 - The institutional and/or licensing committee approving the animal experiments is not indicated.
 - Keywords are missing.

- References are not indicated in the text like demanded by the authors guidelines. The References indicated in the bibliography do not suit the demanded citation style. “

Response: These points have been corrected.

Regarding the mouse model:

- “Fig. 1E: Typing error: It has to be change not "cahnge".

Response: This error was rectified.

- “Fig. 2C/E: Do these 6 representative areas derive from one mammary gland of one animal or from different glands/animals? “

Response: The images show one representative mammary gland of different animals at each time points, therefore representing 8 different animals. However this time course was done in triplicate (3 different animals at each time points). The quantifications were performed using all three mice at each time points. This point has been clarified in the legend.

- “Fig. 2G: Statistical analysis is missing or it has to be indicated, that here was no statistical analysis done.”

Response: Statistical analysis has been performed and is shown on the graph, with significant difference represented by a star. Also details of the statistical analysis performed in each experiment are now indicated in the legend of the figures.

- “Fig. 3C/D: Housekeeping protein is not shown.”

Response: Data in figure 3C is from cell culture media; hence, there is no housekeeping protein, we loaded the same volume of media in each case. Data in figure 3D is a cell-free assay using recombinant IGFBP5 and hence there is no housekeeping protein, we used the same amount of IGFBP-5 in each case. However, in figure 3E, the same decrease in IGFBP-5 was obtained in a cell-based assay and in this case a housekeeping protein could be shown.

- “Fig. 3J: Why is $p=0,01$ declared as not statistically significant?”

Response: We apologize this was an error. The p value from this analysis is 0.06. However, as suggested by the other reviewers, Mann-Whitney-test has now been used for the statistical analysis between the two groups.

- “Fig. 4C and the H&E-section of Fig. 4D do not support the results given in the text. They represent expendable information and maybe should be shifted to the supplementary figures.”

Response: The H&E shown in figure 4C now includes a representation of the different histology found in the PAPP-A tumors at the request of reviewer 1, also phospho-AKT was added for the same reason in figure 4D.

- “Fig. 4G/H: Do the ROI derive from on mammary gland of one animal or from different glands/animals?”

Response: This data arise from the analysis of 6 different ROI from different mice per group. This clarification will be added in the legend.

- *"Fig. 5A: Western Blot image of IGFBP-4 is cloudy compared to IGFBP-5."*

Response: This western was replaced by a better one.

- *"Fig. 6O/P/Q/R: The numbering has to be adapted, to ensure a congruent order in the running text of the results."*

Response: The numbering has been adapted to ensure a smoother running of the text.

Regarding analysis of human samples:

"The quality of the human data stands in contrast to the excellent quality of the transgenic mouse model and lacks to transfer these convincing results to human PABC. Due to the well established methodology it is conceivable to adapt its quality by expanding the number of included patients, by collecting the required additional information on pregnancy and lactation anamnesis and by correcting the current classification of pre-menopausal state as described above.

Unfortunately there is no data available on lactation anamnesis and surprisingly there is no information on the number of pregnancies within the group of parous women.

- The number of patients included is low."

Response: With the help of Drs. Jaffer and Schmidt who are a breast pathologist and a surgeon respectively, we have been able to identify a significant number of additional patients in both groups. While the number of patients in the parous group was 12 in the previous version, it is now 28 in the revised version. In the nulliparous group, the number of patient was 9 in the original version and we now have 17 in the revised version. However, information on lactation anamnesis or number of pregnancies is not available.

"- There was no normality test applied before statistical analysis.

- In cases of small numbers the utilization of median and consequently the utilization of the Mann-Whitney-test for comparison of two groups is recommended"

Response: This point was also raised by reviewer 1 and the statistical analysis have been corrected to Mann-Whitney-test.

Referee #3 (Remarks):

EMM-2015-05692

Title: Lactation opposes pappalysin-1 drive pregnancy-associated breast cancer.

Corresponding author: Doris Germain

This manuscript addresses two clinically important but understudied areas of pregnancy-related breast cancer, namely the mechanisms for increased BC risk following pregnancy and the protective effect of lactation. The breadth of work is ambitious, and the questions fundamental with novel observations. Specifically, using a transgenic model with PAPP-A targeted to the mammary epithelium, the authors identify a role for the protease pappalysin-1 (PAPP-A) in collagen deposition, which in turn, facilitates the proteolysis of IGFBP-4 and -5. In a xenograft model comparing control to PAPP-A overexpressing MCF-7 cells, PAPP-A overexpressing cells form larger tumors with decreased IGFBP-5 levels, implicating the release of IGF as a primary mechanism for the observed increased tumor size. Further, the PAPP-A driven tumors have elevated tumor associated collagen (called TAC3), providing a plausible mechanism for the high metastatic rate of PABC observed in women. Importantly, lactation greater than 2 days in length inhibits PAPP-A driven tumor progression by upregulating expression of the putative PAPP-A inhibitors STC1 and STC2. In the PAPP-A transgenic mice, multiple rounds of pregnancy drive hyperplasia, suggesting PAPP-A as a pregnancy associated oncogene.

The links between PAPP-A, collagen and IGFBP cleavage are novel and provide important

contributions to the breast cancer/breast density/collagen field. The lactation data are also highly novel. Further, the inclusion of collagen TAC assessment in a small cohort of parous and nulliparous breast cancer cases is supportive of the paper's primary hypothesis-and is a commendable addition to the manuscript.

Major Concerns

1. *“The authors rely on overexpression models and do not show a physiologic role for PAPP-A in normal pregnancy, lactation or weaning-induced involution. If the PAPP-A/collagen pathway is causally implicated in the initiation of postpartum breast cancers, upregulation with pregnancy is anticipated.”*

Response: As with other oncogenes it is the gain of function associated with their overexpression that is causally implicated in cancer. We have included data regarding the level of endogenous levels of PAPP-A during the various phases of the mammary gland in supplementary figure 5C. A section was added to the text to address this point. This data indicate that endogenous PAPP-A is not expressed during involution. This result is consistent with the observation that elevated levels of IGFBP-5 are required from normal involution to take place.

2. *“A key point in the paper is that PAPP-A is a pregnancy associated oncogene, as multiple pregnancies result in mammary hyperplasia. The authors clearly state that the MMTV-construct utilized to express PAPP-A is not hormone responsive, thus eliminating the trivial explanation that hyperplasia develops due to hormone regulated transgene expression. However, the authors have not ruled out the role of progesterone (or combined estrogen and progesterone at pregnancy levels) in driving transgene expression. Without these data, it is not possible to conclude that the increase in hyperplasia with pregnancy is not an artifact of the model (albeit, one that yields informative data on collagen regulation). This concern becomes more relevant given that the authors do not demonstrate a role for PAPP-A in normal pregnancy.”*

Response: We have included the data testing the possible role of progesterone. This data is presented in supplementary figure 4A and shows that progesterone alone or in combination with estrogen actually leads to a decrease in expression of PAPP-A.

3. *“Given points 1 & 2, in the manuscript's current form, the data best support a role for PAPP-A in mediating collagen deposition and IGF bioavailability, independent of pregnancy, lactation or involution.”*

Response: Given the answers to points 1 and 2 and the rest of the data presented in the manuscript, we trust this reviewer will agree that the data does support a role of PAPP-A during pregnancy and involution.

4. *“The roles of STC1 and STC2 in mediating the protective effect of lactation are not well developed. Differences in expression/function are expected between the short and full lactation models, but these data are not provided. Currently the data are correlative, and causal roles for STC1 and SCT2 are lacking. The wording of the conclusions should be tempered to reflect the correlative association.”*

Response: Agreed, the wording has been tempered. The previous sentence on p17 : *“Further, the expression of STC 1 and STC2 late during gestation and lactation is able to fully inhibit the basal activity of PAPP-A.”* was replaced by: *“since STC1 and STC2 have been identified as inhibitors of PAPP-A and that they are produced during late pregnancy and lactation, it is tempting to speculate that, the expression of STC 1 and STC2 late during gestation and lactation is able to fully inhibit the basal activity of PAPP-A.”* Also the wording in the titles of section referring to SCT1 have been changed.

Figures:

“Fig 3F compared to Fig 3H, can the authors speculate as to why the addition of collagen to vector-only tumor cells would induce tumor loss? Is this observation robust/replicable?”

Response: This point was also raised by reviewer 1. Our answer is as follow: Collagen has been shown to have both growth-promoting (during involution) and growth-inhibiting (post-involution) properties (Lyons et al., Nat. Med., 2011, Maller et al., J. Cell Science, 2013). Hence, since the experiment performed in figure 3F mimics the post-involution condition, the result suggests that collagen is indeed growth-inhibitory for the control cells. However, when PAPP-A is expressed, collagen enhances its oncogenic function. Discussion of this effect was added to the discussion page 18.

“Fig 5 F, is hyperplasia observed after the 3rd pregnancy in WT mice? “

Response: This data has been added and shows that there is no hyperplasia in a normal gland after 3 pregnancies.

“In many of the figures, the axis font is too small to read”.

Response: The axis fonts have been rectified.

Methods Questions:

“For most of the experiments, it is unclear how many mice were used per group as well as how many replicate studies were performed. Further, error bars are missing from many of the graphs, suggesting absence of true replica data. “

Response: The number of animals has been clarified by adding them to the legends of each figure in addition to the material and method section. These experiments have been done in triplicates.

“For Fig 7, please provide clinical parameters of tumor stage and size, as these parameters could independently influence TAC formation. Also, how many different areas within each tumor were assessed for TACS? What is the intra-tumor variation in TACS? “

Response: Tumor size and grade were added to supplementary figure 6 and indicate no difference between the two groups. 6 different areas (as indicated in methods for patient TACS-3 analysis) were assessed for TACS. Intratumor variability was negligible between the 6 area of individual tumors.

General:

“The cited literature could be updated to include 1) work by others investigating pregnancy-dependent oncogenes using oncogene delivery that is demonstrably hormone independent (Yi Li lab) and literature/discussions on the time frame after pregnancy for which a diagnosis of pregnancy-associated breast cancer is considered, as substantial clinical data supports an extended definition (Albrektsen G., Lambe, M., and Borges, V).”

Response: Agree. Albrektsen G., Lambe, M., and Borges, V. were added to the introduction. Li study added to discussion. However due to word limit, only a brief mention of this literature could be incorporated.

2nd Editorial Decision

14 January 2016

Thank you for the submission of your revised manuscript to EMBO Molecular Medicine. We are sorry that it has taken longer than usual to get back to you on your manuscript. We experienced significant difficulties in securing the re-evaluations. This was compounded with the holiday season and the need to further consult with the reviewers and my colleagues.

As you will see, fundamental concerns remain that preclude publication of the manuscript in EMBO Molecular Medicine.

Reviewer 3 still has concerns related to the unconvincing causal connections, but most crucially, Reviewer 2 notes that you have not fully addressed the original issue of the very low statistical power of your animal experimentation.

As mentioned above, I further consulted with the Reviewers during our cross-commenting exercise. One reviewer unfortunately came to agree with Reviewer #2's assessment, expressing regret for the oversight. Indeed, s/he noted that the issues raised extend even further: in many experiments, the total number of mice is not even mentioned. Also the fact that for most of the mouse experiments the number of mice is 3/group, generates concern with respect to the statistical power regardless of the t-test.

In conclusion, it was agreed that what is fundamentally lacking in the manuscript are, for example, a few key-experiments with a large-enough sample size to demonstrate that a) the values compared are normally distributed and b) one can expect enough power in experiments with smaller sample sizes.

I hope that you understand that, also considering our policy to allow a single major round of revision only (except for minor amendments), albeit not light-heartedly I have no choice but to return the manuscript to you at this stage so that you may choose an alternative venue for your work

Given our (and the Reviewers') interest in your work, we would have no objection to consider a new manuscript if at some time in the near future you have obtained data that validate the key findings with adequate animal numbers and with careful revision of the statistics, in addition to adequately addressing the concerns of Reviewer 3.

***** Reviewer's comments *****

Referee #1 (Remarks):

The authors have responded to each point raised by the reviewers, thereby providing additional results and further strengthening the conclusions drawn.
The authors are to be commended for their diligence in responding to all of the reviewers.

Referee #2 (Comments on Novelty/Model System):

The use of n=2, 3 or 4 mice/group is not acceptable. Besides, the choice of graphic is wrong. Showing mean +/- SEM implies that the samples are normally distributed within a group. This fact can not be even checked with n=3/group. The statistical tests used to compare the groups are also wrong. It is a pity that having such a nice hypothesis do not test it correctly.

Referee #2 (Remarks):

The use of n=2, 3 or 4 mice/group is not acceptable. Besides, the choice of graphic is wrong. Showing mean +/- SEM implies that the samples are normally distributed within a group. This fact can not be even checked with n=3/group. The statistical tests used to compare the groups are also wrong.
It is unclear whether the conclusions drawn from few experimental samples are correct.

Referee #3 (Remarks):

The authors have developed convincing and provocative data demonstrating that PAPP-A increases collagen deposition during weaning-induced, murine mammary gland involution. The authors also describe a murine model that, for the first time, models the human epidemiologic data identifying lactation as a risk factor for breast cancer, and for these reasons this work is timely and important. Their data support a mechanism that is consistent with PAPP-A cleavage of IGFBP-4 and-5, with release of IGF, leading to delayed involution and collagen deposition. The mechanistic link between lactation and inhibition of PAPP-A activity remains to be determined.

My initial concerns that the MMTV transgene was upregulated during pregnancy or by hormones of pregnancy have been adequately addressed. However, there are still several areas where the clarity of the paper can be improved. For example, most breast cancer researchers using mouse MMTV

models are not as familiar as the authors with this particular non-hormone responsive MMTV-promoter, and given the importance of the hormone-controls for their data interpretation, I suggest Fig 1Sa, b & c be moved to the main body of the paper. Further, much of the lengthy discussion of the non-hormone responsive MMTV construct that was added on page 7 of the 'Results' section could be consolidated with the information presented on page 12 of 'Results' and moved to the 'Discussion' section.

The increase in number of human breast cancer tissues assessed for PAPP-A and collagen 'curvlet angles' also improves the manuscript, even in the absence of lactation data, which, as the authors correctly point out, are very difficult to obtain from clinical chart review. However, the human data are associative only, leading to the overall concern that several conclusions in the paper are overstated or ambiguous as to whether the conclusions are specific to this animal model or generalized to women. For example, in the Discussion, concerning the 2002 publication from the Collaborative Group on Hormonal Factors in Breast (that provides epidemiologic data suggesting increased lactation might reduce cumulative risk of breast cancer by half), the authors' state "Our results are consistent with these observations and, to our knowledge, offer the first molecular mechanism that explains the protective effect of lactation."

Concerns about the lack of a basis to extend conclusions from the current study to imply causality in human pregnancy-associated breast cancer stem in part from the authors own data showing that the collagen TAC-3 signature is specific to PAPP-A driven murine mammary tumors, but not MMTV-HER2 or MMTV-WNT tumors. Thus, their murine data indicate that lactation would be preferentially protective in PAPP-A driven tumors in women, but the human data suggest the vast majority of parous patients have elevated TAC-3s, independent of driver mutation or lactation history. Importantly, the demonstration of elevated collagen in breast cancers of pre-menopausal, parous women does not directly implicate PAPP-A or lack of lactation. These concerns are highlighted by the fact that previous studies published by co-author Keely in collaboration with Schedin, have already demonstrated increased collagen in postpartum DCIS, with elevated collagen levels in human postpartum PABC predicted based on a Cox-2 collagen feedback loop identified in mice.

Specifically, for the data presented by Takabatake et al, only one of the four criteria to establish a causal relationship between PAPP-A, pregnancy associated breast cancer and lactation has been solidly met; overexpression of PAPP-A can drive an aggressive breast cancer after pregnancy, and this phenotype can be mitigated by lactation. Evidence that PAPP-A is normally regulated by pregnancy/lactation in mice or women (the first requirement of Koch's postulate) is lacking or null. Rather, than offering " ...the first molecular mechanism that explains the protective effect of lactation", data presented are hypothesis generating and not directly translatable to women. I recommend that wording throughout the text be tempered to reflect 1) the fact that the primary data are obtained from a transgenic model and overexpressing cell lines, 2) that evidence for a role for PAPP-A in normal pregnancy/lactation are lacking, and 3) that a discrepancy exists between predictions of the murine model (PAPP-A specific phenotype) and human data showing most parous BC cases have TACS-3 collagen.

Other comments/questions:

1. Data showing STC1 and STC2 elevated during pregnancy and lactation are weak, triplicate data should be shown as well as quantitation of these data.
2. Long lactation is reported as more than 2 weeks, but the actual range is 21-24 days. I would change long lactation to state '3 weeks or more', as this better reflects the study design and separates your groups more convincingly.
3. The lack of intra-tumor variability in TAC-3 scores is surprising given the heterogeneity of human breast cancer.
4. Fig 6K and 6L could be moved to supplement.
5. Please clarify the meaning of "saturation of PAPP-A" (page 21).

Please find enclosed the second revision of our manuscript entitled “Lactation opposes Pappalysin-1 driven pregnancy-associated breast cancer”, that we are re-submitting for your consideration for publication in EMBO Molecular Medicine.

The major remaining concern related to the number of mice and that “*n=2, 3 or 4 is not acceptable*”. In response to your last letter, *we have increased the number of mice* for experiments where that was possible. Namely, the number of mice was 4 in figure 4A and it is now 7. In addition, the number of mice was 2 in figure 4E and it is now 5. Therefore, there are no experiment with $n=2$ anymore. For experiments where $n=3$, these fall into two categories 1) xenografts and 2) time course in transgenic mice.

1) For the xenografts, only experiments in figure 6 are using $n=3$ with 2 tumors per mouse (6 tumors total). Our biostatistician has suggested that for all xenografts experiments, we make the average of the 2 tumors per mouse and plot the difference between the two groups. As there is no overlap between the measurements of the two groups (now shown in supplementary figures), the use of 3 mice does offer statistical power. Importantly, figure 6B is a repeat of experiment 3I where 4 mice (8 tumors) were used and we obtain the same result. For figure 6C and D, we show no statistical difference between the two groups and concluded that lactation abolish the difference between the two groups. While this experiment could still be criticize for using $n=3$, I would like to point out that the conclusion that lactation is protective does not rely only on panels 6C and 6D. Experiment of the involution time course after a long lactation in transgenic mice (Fig 6E) and the frequency of spontaneous tumors in mice with short and long lactation in figure 6A also support this conclusion. I calculated that if all mice are included, we *analyzed 68 mice to reach this conclusion*.

2) For the time course experiments in transgenic mice. Each time point used $n=3$ and there was 4 different time points for a total of 12 mice. We also had performed 3 independent measurements per mouse. Our biostatistician suggested we make the mean of the 3 measurements for each mouse and compared the two groups (non-transgenic and transgenic). Again we found no overlap between the two groups indicating that $n=3$ does provide statistical power. I would like to add that this *statistical significance is reproducible for more than one time point*.

This was true for both the involution time course and the pregnancy time course. New graphs are shown. Since reviewer 2 simply stated that “the graphs are wrong”, without details, we included both type of graphs and we can eliminate some if you or the reviewer prefers.

For the experiments where $n=4$, this is in figure 3, xenografts with 2 tumors per mouse. We applied the same statistical analysis (average the 2 tumors per mouse) as used for experiment where $n=3$ and found no overlap between the groups, again indicating that the number of mice used offer statistical power to detect a difference between the groups.

The remaining concerns of reviewer 3 have been addressed: some of the data that was shown as supplementary figures has been moved to the main figures, we added a western of a triplicate of STC1 and 2 during the various phases of the mammary gland development. The source file was added as well. We also changed the text to tone down the direct correlation of our findings in mice to human PABC. In addition, the comment of reviewer 3 relating to the data on MMTV-erbB2 and Wnt made us realized that as we do not know the pregnancy history of these mice, they cannot be compared to the MMTV-PAPP-A, so this data was removed as it does not affect the conclusion of this figure.

I would like to point out that reviewer 3 states that “evidence that PAPP-A is normally regulated by pregnancy/lactation is lacking”. This reviewer had made that point in the first round of review and we had responded by adding an analysis of endogenous PAPP-A in all phases of mammary glands and added an entire section to address this point in the text and as well as in the model in figure 8. So perhaps reviewer 3 missed it but we have addressed this concern already in the first revision.

Lastly, the biostatistician, Dr. Mandeli, was added as an author. A detailed list of the changes that were made in responses to specific concerns is shown below. Therefore, in light of these additional changes, I trust you will find our manuscript appropriate for publication in EMBO Molecular Medicine.

Response to Referees

Referee #1 (Remarks):

“The authors have responded to each point raised by the reviewers, thereby providing additional results and further strengthening the conclusions drawn. The authors are to be commended for their diligence in responding to all of the reviewers.”

Response: Thank you

Referee #2 (Comments on Novelty/Model System):

“The use of $n=2, 3$ or 4 mice/group is not acceptable. Besides, the choice of graphic is wrong. Showing mean \pm SEM implies that the samples are normally distributed within a group. This fact cannot be even checked with $n=3$ /group. The statistical tests used to compare the groups are also wrong. It is a pity that having such a nice hypothesis do not test it correctly.”

Referee #2 (Remarks):

The use of $n=2, 3$ or 4 mice/group is not acceptable. Besides, the choice of graphic is wrong. Showing mean \pm SEM implies that the samples are normally distributed within a group. This fact cannot be even checked with $n=3$ /group. The statistical tests used to compare the groups are also wrong. It is unclear whether the conclusions drawn from few experimental samples are correct.”

Response: The easiest way to answer this concern is to address the statistical analysis of each type of experiment.

Statistical analysis of experiments in transgenic mice:

Figure 2B: In this experiment 3 mice per time points were used and 4 time points per genotype were analyzed for a total of 12 mice per group. In the previous version, quantification of each sample was performed using 3 different areas of the same slide, which we showed as 9 measurements (3 area \times 3 mice) per time point per genotype. In the revised version, we used the mean of the quantification of 3 area/slide as one measurement, allowing better precision of measurements on experimental units (mice). The experimental units being mice and not individual determinations on mice. Using t-test to compare wild type and transgenic mice at each time point, we found statistical significance at day 6 and 12. The new bar graph is now shown in supplementary figure 2A and the graph showing the distribution of the measurements between the two groups has been shown in figure 2B. Importantly, this graph shows that there is no overlap between the measurements between the NT group and the PAPP-A group at day 6 and 12, and a large difference (more than 2 standard deviations) between the means of the two groups, indicating that 3 mice per time point has sufficient statistical power to detect a difference. Also please note that statistical significance is observed at more than one time point.

Figure 2D: Same changes as 2B were made and again clearly no overlap between measurements between groups.

Figure 4B: The number of mice was increased from 4 to 7.

Figure 4E, F, G and H: In this experiment we aimed at comparing the rate of TACS3 in three different mouse models of breast cancer and while several areas were analyzed, these areas were from only 2 tumors in the MMTV-ErbB2 and Wnt mice. However, in light of the comments from reviewer 3, we realized that we actually do not know the pregnancy history of these ErbB2 and Wnt mice and therefore we cannot compare the PAPP-A tumors to these other tumors. Therefore the data related to MMTV-ErbB2 and Wnt was removed. Instead we increased the number of PAPP-A tumors analyzed from 2 to 5. The conclusion of this analysis is that TACS3 can be found in the PAPP-A tumors (new figures F and G).

Figure 5D: This experiment is also a time course using 3 mice and 3 different time points during pregnancy. As for figure 2B, the previous version used 3 different areas for all 3 mice for a total of 9 measurements. This was changed to the mean quantification of 3 areas per mouse and t-test. This analysis confirmed statistical significance at day 9 and 12. Further, graph of the distribution of

the measurements between groups shows a large difference. This new graph is shown in figure 5D and the bar graph is shown in supplementary figure 2C.

Statistical analysis of experiments in xenografts:

Figure 3F, H, and I: In these experiments, 4 mice per group were used but each mouse had 2 tumors, therefore in the previous version, we had performed the statistical analysis using 8 tumors per group. In the revised version, we used the average volume of the 2 tumors/mouse as a way to obtain a measurement of average tumor volume for each mouse. We have performed t-tests at the final time point of each experiment and confirmed statistical significance in 3F and 3I but no significance in 3H. Again graphs of the distributions (supplementary figure 5) of measurements between groups show a large difference in the means between groups and no overlap between measurements in the groups where significance is found.

Figure 3J: In these experiments, 3 mice per group were used with 1-2 tumors per mouse (5 tumors total), therefore we had performed the statistical analysis using 5 tumors total per group. In the revised version, we used the average volume of the 1-2 tumors/mouse as a way to improve precision in the measurements for each mouse. We have performed a t-test and confirmed statistical significance of the difference in growth rate (represented by the slope of the tumor volume graph) before PQ401 treatment between control and PAPP-A tumors, but not after PQ401 treatment. Once again, graphs of the distributions (supplementary figure 5) show a large difference between means and no overlap where significance is found.

Figure 6B, C, D: In these experiments, 3 mice per group were used with 2 tumors per mouse, therefore we had performed again the analysis on 6 tumors. In the revised version, we correct this analysis by using the average volume of the 2 tumors and again found significance in 6B with a very large difference (3 standard deviations) between the means of the groups with no overlap (Supplementary figure 6). Importantly, data in figure 6B is a repeat of experiment in figure 3I so this is a duplicate of the same experiment, which confirms the result of 3I. Therefore, it can be argued that this result is based on the analysis of 7 mice total. To address the concern that our conclusions are based on too few mice, the conclusion that lactation is inhibitory from panel 6C and D is supported by an independent experiment done in transgenic mice in figure 6E, where 12 mice were used. Further, in panel 6A we show that only mice with long lactation are protected and this analysis is done in 12 and 14 mice respectively. Therefore, we trust that the reviewer will acknowledge that our conclusion is not drawn simply on a single experiment using 3 mice per group but on several experiments (6A, B, C, D and E) combining the analysis of 68 mice.

Referee #3 (Remarks):

“The authors have developed convincing and provocative data demonstrating that PAPP-A increases collagen deposition during weaning-induced, murine mammary gland involution. The authors also describe a murine model that, for the first time, models the human epidemiologic data identifying lactation as a risk factor for breast cancer, and for these reasons this work is timely and important. Their data support a mechanism that is consistent with PAPP-A cleavage of IGFBP-4 and -5, with release of IGF, leading to delayed involution and collagen deposition. The mechanistic link between lactation and inhibition of PAPP-A activity remains to be determined. My initial concerns that the MMTV transgene was upregulated during pregnancy or by hormones of pregnancy have been adequately addressed. However, there are still several areas where the clarity of the paper can be improved. For example, most breast cancer researchers using mouse MMTV models are not as familiar as the authors with this particular non-hormone responsive MMTV promoter, and given the importance of the hormone-controls for their data interpretation, I suggest Fig 1Sa, b & c be moved to the main body of the paper. Further, much of the lengthy discussion of the non-hormone responsive MMTV construct that was added on page 7 of the 'Results' section could be consolidated with the information presented on page 12 of 'Results' and moved to the 'Discussion' section.”

Response: Fig 1Sa, b and c have been moved to figure 1 as Fig 1A, B and C. Because of space issue the previous Fig 1A was moved as supplementary figure 1A. Data regarding the MMTV promoter has been moved to figure 4E. We also consolidated the discussion of the MMTV promoter on page 7.

"The increase in number of human breast cancer tissues assessed for PAPP-A and collagen 'curvlet angles' also improves the manuscript, even in the absence of lactation data, which, as the authors correctly point out, are very difficult to obtain from clinical chart review. However, the human data are associative only, leading to the overall concern that several conclusions in the paper are overstated or ambiguous as to whether the conclusions are specific to this animal model or generalized to women. For example, in the Discussion, concerning the 2002 publication from the Collaborative Group on Hormonal Factors in Breast (that provides epidemiologic data suggesting increased lactation might reduce cumulative risk of breast cancer by half), the authors' state "Our results are consistent with these observations and, to our knowledge, offer the first molecular mechanism that explains the protective effect of lactation."

Response: We have toned down by changing the sentence "Our results are consistent with these observations and, to our knowledge, offer the first molecular mechanism that explains the protective effect of lactation" by the sentence "Our results are in agreement with these observations and offer a potential mechanism that contributes to the protective effect of lactation."

"Concerns about the lack of a basis to extend conclusions from the current study to imply causality in human pregnancy-associated breast cancer stem in part from the authors own data showing that the collagen TAC-3 signature is specific to PAPP-A driven murine mammary tumors, but not MMTV-HER2 or MMTVWNT tumors. Thus, their murine data indicate that lactation would be preferentially protective in PAPP-A driven tumors in women, but the human data suggest the vast majority of parous patients have elevated TAC-3s, independent of driver mutation or lactation history. Importantly, the demonstration of elevated collagen in breast cancers of pre-menopausal, parous women does not directly implicate PAPP-A or lack of lactation. These concerns are highlighted by the fact that previous studies published by co-author Keely in collaboration with Schedin, have already demonstrated increased collagen in postpartum DCIS, with elevated collagen levels in human postpartum PABC predicted based on a Cox-2 collagen feedback loop identified in mice."

Response: We did not intend to imply that TACS3 only occur in the PAPP-A driven tumors. We simply suggested that since collagen deposition is higher during involution of the PAPP-A mice than in wt mice, TACS3 may be higher than in other models. We do know that collagen is increased in postpartum DCIS. So this point is well taken and the text was modified to clarify this aspect. More importantly, this comment made us realize that we do not actually know the pregnancy history of the ErbB2 and Wnt mice. Therefore, since these tumors may have arisen in virgin mice, we cannot actually make this comparison so the data was removed. Rather, we simplified to say that we can detect TACS3 in the PAPP-A tumors.

"Specifically, for the data presented by Takabatake et al, only one of the four criteria to establish a causal relationship between PAPP-A, pregnancy associated breast cancer and lactation has been solidly met; overexpression of PAPP-A can drive an aggressive breast cancer after pregnancy, and this phenotype can be mitigated by lactation. Evidence that PAPP-A is normally regulated by pregnancy/lactation in mice or women (the first requirement of Koch's postulate) is lacking or null. Rather, than offering " ...the first molecular mechanism that explains the protective effect of lactation", data presented are hypothesis generating and not directly translatable to women. I recommend that wording throughout the text be tempered to reflect 1) the fact that the primary data are obtained from a transgenic model and overexpressing cell lines, 2) that evidence for a role for PAPP-A in normal pregnancy/lactation are lacking, and 3) that a discrepancy exists between predictions of the murine model (PAPP-A specific phenotype) and human data showing most parous BC cases have TACS-3 collagen."

Response: The text has been tempered. Of note however, we did respond to this concern after the first round of review by including a section about the normal regulation of PAPP-A during pregnancy. Please see page 17.

Other comments/questions:

1. Data showing STC1 and STC2 elevated during pregnancy and lactation are weak, triplicate data should be shown as well as quantitation of these data.

Response: This western was performed and replaced in the new version.

2. Long lactation is reported as more than 2 weeks, but the actual range is 21-24 days. I would change long lactation to state '3 weeks or more', as this better reflects the study design and separates your groups more convincingly.

Response: This was corrected

3. The lack of intra-tumor variability in TAC-3 scores is surprising given the heterogeneity of human breast cancer.

Response: We have now included supplementary figure 8 to show the intra-tumor variability of TACS-3 between the 6 areas from individual tumors represented by the standard error of the mean.

4. Fig 6K and 6L could be moved to supplement

Response: This was corrected and can be found in supplementary figure 6B, C.

5. Please clarify the meaning of "saturation of PAPP-A" (page 21).

Response: We changed "saturation" for "inactivation".

3rd Editorial Decision

10 February 2016

Thank you for the submission of a revised version of former manuscript EMM-2015-05692 to EMBO Molecular Medicine

Based on an internal discussion at the editorial level, we are now satisfied that you have adequately addressed the previous concerns on statistical treatment and power, in addition to providing appropriate responses to Reviewer 3's remaining requests.

I am pleased to inform you that we will be able to accept your manuscript pending the following final amendments:

Corresponding Author Name: Doris Germain

Manuscript Number: EMM-2015-05692